# Evaluation of long-term carbon dynamics in a drained forested peatland using the ForSAFE-Peat Model.

Daniel Escobar[1,2], Stefano Manzoni[1], Jeimar Tapasco[2], Patrik Vestin[3], Salim Belyazid[1]

[1] Department of Physical Geography and Bolin Centre for Climate Research, Stockholm University, SE-106 91 Stockholm, Sweden.

[2] Climate Action, Alliance of Bioversity International and the International Centre for Tropical Agriculture (CIAT), Palmira 763537, Colombia.

[3] Department of Physical Geography and Ecosystem Science, Lund University, SE-22362 Lund, Sweden.

*Correspondence to*: Daniel Escobar (daniel.escobar@natgeo.su.se)

**Abstract.** Management of drained forested peatlands has important implications for carbon budgets, but contrasting views exist on its effects on climate. This study utilised the dynamic ecosystem model ForSAFE-Peat to simulate biogeochemical dynamics over two complete forest rotations (1951-2088) in a nutrient-rich drained peatland afforested with Norway spruce (*Picea abies*) in southwest Sweden. Model simulations aligned well with observed groundwater levels ($R^2 = 0.78$) and soil temperatures ($R^2 \geq 0.76$), and captured seasonal and annual net ecosystem production patterns, although daily variability was not always well represented. Simulated carbon exchanges (a positive sign indicates gains and a negative sign indicates losses) were analysed considering different system boundaries (soil, ecosystem, and ecosystem plus the fate of harvested wood products, named ecosystem+HWP) using the net carbon balance (NCB) and the integrated carbon storage (ICS) metrics. Model results indicated negative NCB and ICS across all system boundaries, except for a positive NCB calculated by the end of the simulation at the ecosystem+HWP level. The soil exhibited persistent carbon losses primarily driven by peat decomposition. At the ecosystem level, net carbon losses were reduced as forest growth partially offset soil losses until harvesting. NCB was positive ($2307$ $g_C$ $m^{-2}_{soil}$) at the ecosystem +HWP level due to the slow decay of harvested wood products, but ICS was negative ($-0.59 \times 10^6$ $g_C$ yr $m^{-2}_{soil}$) due to the large initial carbon losses. This study highlights the importance of system boundary selection and temporal dynamics in assessing the carbon balance of forested drained peatlands.

## 1 Introduction

Atmospheric greenhouse gas (GHG) concentrations consistent with the Paris Agreement's long-term temperature goal require ambitious carbon removals during this century (Rogelj et al., 2018). Land management practices can lead to net removals or net exports depending on several controlling factors, often hard to quantify and generalise (Crusius, 2020; Guenther et al., 2020; Krause et al., 2020; Seddon et al., 2020). This problem is particularly acute in peatlands, as they are large carbon stores and very sensitive to land management.

Forestry on drained peatlands is a widespread land management practice in the northern hemisphere, covering approximately 15 million hectares, and it has important implications for carbon budgets (Leifeld et al., 2019). This practice is widespread in Fennoscandia, spanning around 5.7 million hectares in Finland and 1.5 million hectares in Sweden (Vasander et al., 2003). Drainage leads to important changes in the carbon dynamics of these systems (Ojanen and Minkkinen, 2019). Lowering the

water table promotes forest growth and, subsequently, carbon accumulation in living biomass and decreases methane emissions (Escobar et al., 2022). Nonetheless, higher soil oxygen content associated with lowering the water table promotes decomposition, potentially leading to substantial carbon emissions from peat soils (He et al., 2016). According to Jauhiainen et al. (2023), the soil carbon balance, calculated as the difference between litter inputs and heterotrophic respiration, commonly shows soil carbon losses for drained forested peatlands at northern latitudes, ranging from 21 and 261 $g_C$ $m^{-2}$ $yr^{-1}$ depending

on climate and nutrient status.

Restoration of water table levels and wetland vegetation has been proposed to meet Paris Agreement targets (Guenther et al., 2020; Tanneberger et al., 2021). Several efforts to restore peatlands are underway. For example, the EU Nature Restoration Law has proposed specific area targets for peatland rewetting (Noebel, 2023). Drained peatlands restored through rewetting exhibit long-lasting differences regarding hydrological and ecological dynamics compared to their pre-drainage status

(Kreyling et al., 2021). However, restoration seems capable of reducing soil carbon losses in these systems (Darusman et al., 2023; Escobar et al., 2022).

While restoration through rewetting holds promise for mitigating climate change, its effectiveness remains a subject of debate due to different views about the effects on climate caused by drained forested peatlands at northern latitudes (Kasimir et al., 2018; Meyer et al., 2013; Ojanen and Minkkinen, 2020). Whether all types of drained peatlands consistently lose soil carbon

is still an open question due to contrasting results from field measurements (Butlers et al., 2024; Hermans et al., 2022; Meyer et al., 2013; Minkkinen et al., 2018). Additionally, disagreement persists regarding the appropriate boundaries for analysing these systems, specifically whether carbon accumulated in harvested tree biomass should be included in the carbon budgets to estimate climate impacts in a timeframe relevant to climate change mitigation (Kasimir et al., 2018; Ojanen and Minkkinen, 2020).

The importance of the tree biomass components is clear from net ecosystem production (NEP) measurements performed with the eddy covariance technique, which indicate a persistent carbon sink in drained forested peatlands despite high soil carbon losses (Korkiakoski et al., 2019; Meyer et al., 2013; Tong et al., 2024). It has been recognised that in cases of persistent and large soil carbon losses, compensation through forest carbon uptake is limited because the tree component has a maximum carbon storage capacity lower than the carbon stocks of typical peat soil. The magnitude and extent of this compensation are

likely sensitive to how harvested wood products (HWP) are accounted for. When considering HWP, post-harvesting periods are of special relevance, suggesting that it is necessary to analyse carbon dynamics over more than one forest rotation to understand the trade-off between tree biomass carbon and soil carbon. This shows how differences in system boundary definition, meaning considering the carbon balance within the soil, ecosystem, or the ecosystem plus the fate of HWP, may lead to contrasting results.

Furthermore, due to tree carbon uptake compensation of soil carbon losses, the effects on the climate of these systems might be greatly affected by how the forest stand is managed (Tong et al., 2024), which adds uncertainties to the estimated carbon budgets. Indeed, a large area of drained forested peatlands will likely undergo conventional forest management in the next few decades (Lehtonen et al., 2023). Field-based measurements of carbon balances have shown high temporal variability due to high sensitivity to nutrient status, forest stand characteristics, water table level and temperature (Korkiakoski et al., 2023;

Mamkin et al., 2023) .Adding to these uncertainties, measurements are usually performed during short periods (Escobar et al., 2022) that do not correspond to the long cycles of conventional forestry. To complement short-term measurements, dynamic ecosystem models can provide simulation data about carbon dynamics representative of long periods that can be further analysed under different system boundaries (Minkkinen et al., 2018).

Here, we introduce the dynamic ecosystem model ForSAFE-Peat and use it to analyse long-term carbon dynamics in a drained

forested peatland. ForSAFE-Peat builds on previous models of carbon dynamics in coniferous forest and peat soils. It simulates plant dynamics as a big leaf model where photosynthesis is a function of foliar nitrogen content as in the PnET model (Aber and Federer, 1992). This representation has been widely use to study managed coniferous forest in northern latitudes (Belyazid et al., 2011; Belyazid and Zanchi, 2019; de Bruijn et al., 2014; Gustafson et al., 2020). ForSAFE-Peat simulates the soil as a set of layers that can expand or contract due to soil organic matter content changes, similar to peat development models like

HPM (Frolking et al., 2010). Soil organic matter is represented by several compartments, including litter that, during decomposition, provides carbon and nutrient inputs to peat pools, resembling approaches like the one implemented in Yasso07 (Didion et al., 2014). This allows a simple representation of litter quality and peat. Decomposition is described as a first-order exponential decay process where the peat decomposition rate constant is the same used to evaluate future carbon dynamics of northern peatlands by land-surface models such as ORCHIDEE (Qiu et al., 2018) and LPJ-GUESS (Chaudhary et al., 2022).

By building on existing state-of-the-art models, ForSAFE-Peat is a suitable tool for exploring carbon dynamics in peatland systems and critically examining commonly used methods for their representation.

In this study, we used the ForSAFE-Peat model to conduct a long-term simulation spanning two complete forest rotations in a well-studied drained forested peatland in southwest Sweden, utilising primarily pre-calibrated parameters. Model outputs were analysed to represent various system boundaries and different metrics were applied to evaluate carbon exchanges across these

boundaries. While acknowledging the potential significance of $N_2O$ emissions in drained fertile peatlands (Jauhiainen et al., 2023), we focused on carbon dynamics. Consequently, we explore the following two questions:

i.   How well does ForSAFE-Peat reproduce field-based observations related to carbon dynamics in a northern drained forested peatland?

ii.  How do patterns of modelled carbon exchange vary across different system boundaries in a northern drained forested

peatland?

## 2 Methods

We modified the forest ecosystem model ForSAFE (Wallman et al., 2005; Zanchi et al., 2021b) to better describe prominent processes in peat soils. We then used the modified model ForSAFE-Peat to simulate biogeochemical dynamics encompassing two complete forest rotations in a drained nutrient-rich peatland planted with Norway spruce (*Picea abies*). Site conditions
were typical of drained forested peatlands in southwest Sweden under conventional forestry management practices.

For the first question, we compared model outputs to field measurements performed in an intensively monitored site using goodness of fit indicators. For the second question, we used model outputs to quantify two carbon exchange metrics under different system boundaries and analysed their evolution throughout time.

### 2.1 Model description

ForSAFE-peat simulates daily biogeochemical dynamics building upon the established ForSAFE model (Wallman et al., 2005; Yu et al., 2018; Zanchi et al., 2021b). This process-based and compartmental model tracks carbon, water and nutrient flows throughout a forest stand ecosystem. A detailed description of the model and its mathematical formulation can be found in Supplementary Information 1; here, we only provide a short summary.

The model simulates daily photosynthesis as a function of photosynthetically active radiation, temperature, leaf area, foliar
nitrogen content, water availability, and atmospheric $CO_2$ concentration. Photosynthesised carbon and assimilated nutrients are initially allocated within five labile compartments before entering four specific plant compartments (leaves, branches, wood, and roots). Carbon and nutrients are either harvested or returned to the soil through litterfall for further cycling through decomposition. Woody residues associated with thinning and harvest are allocated to an intermediate compartment of deadwood before entering the soil as litter.

Soil is represented by layers defined by the user, and each layer thickness is allowed to vary during the simulation based on the amount of organic matter it holds while porosity remains constant. Heat is transported vertically according to the heat equation adapted for peat soils. Downward water movement is driven by gravity and modulated by soil hydrological properties, while plants influence water uptake through transpiration. Additionally, specific layers can exchange water horizontally, simulating the impact of ditching on hydrological processes within the peatland. The ditch function is simulated by setting an
initial drainage depth. Layers above this depth experience lateral outflow when water content exceeds field capacity, with outflow regulated by the layer's hydraulic conductivity and width, as described in Zanchi et al. (2021b). The drainage depth adjusts dynamically with changes in the soil profile; when the soil profile height is reduced due to net losses of soil organic matter, the ditch depth is also reduced by the same magnitude.

Organic matter within the soil is divided among four solid compartments (easily decomposable compounds, cellulose, lignin
and peat) that are decomposed at different rates according to first-order kinetics modified by temperature, moisture, and pH. This process releases dissolved organic compounds (dissolved organic carbon, dissolved organic nitrogen and $CH_4$) and mineral compounds ($CO_2$, $NH_4^+$, $Mg^+$, $K^+$ and $Ca^+$) into the soil solution. Mineral weathering and atmospheric deposition

contribute compounds such as sulphate ($SO_4^-$), nitrogen ions ($NH_4^+$, $NO_3^-$), base cations ($Mg^+$, $K^+$, $Ca^+$), chloride ($Cl^-$), sodium ($Na^+$) and aluminium ($Al^+$) to the soil solution. Atmospheric deposition, influenced by historical and local conditions, is a direct input of these compounds and ions. At the same time, mineral weathering depends on the mineral content, reducing its significance in organic soils with lower mineral availability. Additionally, ion exchange processes regulate the availability of these compounds through adsorption or desorption. Leaching, driven by water exports, removes compounds from the soil solution.

Mass balance equations that account for gas-water partitioning, diffusion, water transport, plant uptake and chemical transformations are used to track the concentration of these elements in the soil. Soil solution pH is then calculated based on the acid-neutralizing capacity of the soil solution.

The model tracks the fate of carbon within the harvested biomass extracted from the site by allocating it into three compartments (fuel, fibre and hardwood products) whose decay is simulated through first-order kinetics and has no feedback on other parts of the model.

## 2.2 Site and scenario description

We simulated two forest rotations over the period from the beginning of 1951 to the end of 2088 at a drained afforested peatland located at Skogaryd Research Station (Klemedtsson et al., 2015) in the southwest of Sweden (58°23'N, 12°09'E). This site experiences a hemiboreal climate, has nitrogen-rich peat soil, features an effective drainage system, and is managed under conventional forestry practices. Originally an open fen valley, the site was drained in the late 19th century for agriculture before being converted to forestry in 1951. The ditch network forms a grid-like pattern, with the main ditch running north to south for 0.8 km, draining into Lake Skottenesjön. Smaller parallel ditches are spaced at varying distances. Until clear-cutting in 2019, the site was dominated by Norway spruce (*Picea abies*). The area affected by clear-cutting covered approximately 0.16 km², with logging debris left on most of the site (Figure 1). Norway spruce was replanted on 2/3 of the site following clear-cutting. In 2022, a barrier was constructed in the main ditch to raise the water level in the northern third of the site. Visual inspections revealed that vegetation cover increased in the years following clear-cutting. By 2022, much of the site remained covered by logging residues, while grasses and sedges, particularly in areas without logging debris, reached heights of 90 cm in the middle of the summer.

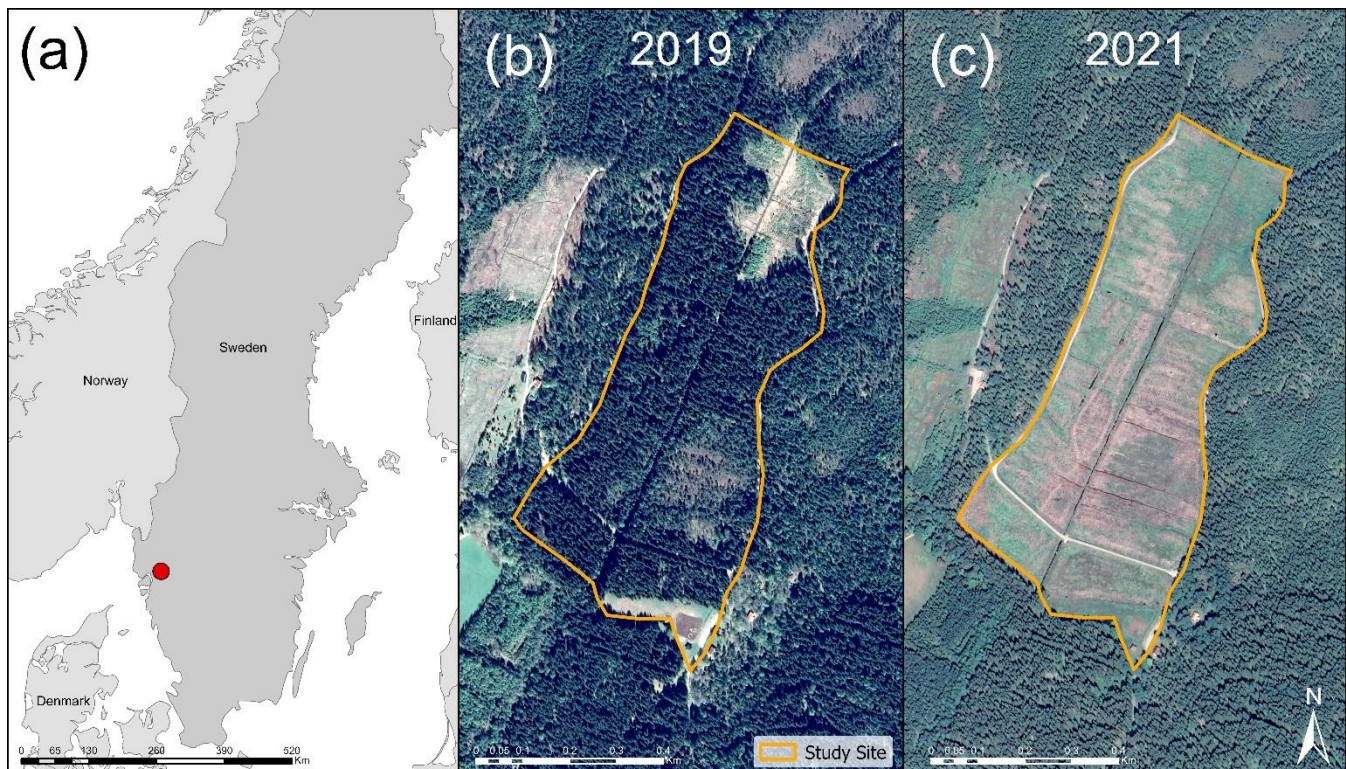

Figure 1. (a) Location of the Skogaryd Research Station (58°23'N, 12°09'E) in southern Sweden, marked with a red circle. (b) Satellite image of the study site in 2019, showing a predominantly coniferous forest. (c) Satellite image from 2021 after clear-cutting, revealing an extensive drainage network. Satellite images were obtained from Google Earth Pro.

The model used daily mean meteorological data (1951 to 2023) from the Swedish Meteorological and Hydrological Institute (SMHI) Uddevalla (58°36'N, 11°93'E) and Vänersborg (58°35'N, 12°35'E) stations, both located approximately 12 km from the site. Future climate data (2023 to 2088) were obtained from projections for forest sites under the CLEO research program (Munthe et al., 2016). Climate projections were downscaled from regional projections based on ECHAM and HADLEY climate models under RCP 6.0 as in Zanchi et al. (2021a). RCP 6.0 represents a medium stabilisation pathway, where greenhouse gas emissions peak around 2080 and decline thereafter, reflecting a future with moderate climate change mitigation efforts. Yearly atmospheric deposition was derived from the MATCH model simulation (Engardt and Langner, 2013; Munthe et al., 2016) and scaled based on daily precipitation. Climate data used as an input for the simulations can be seen in Figure 2.

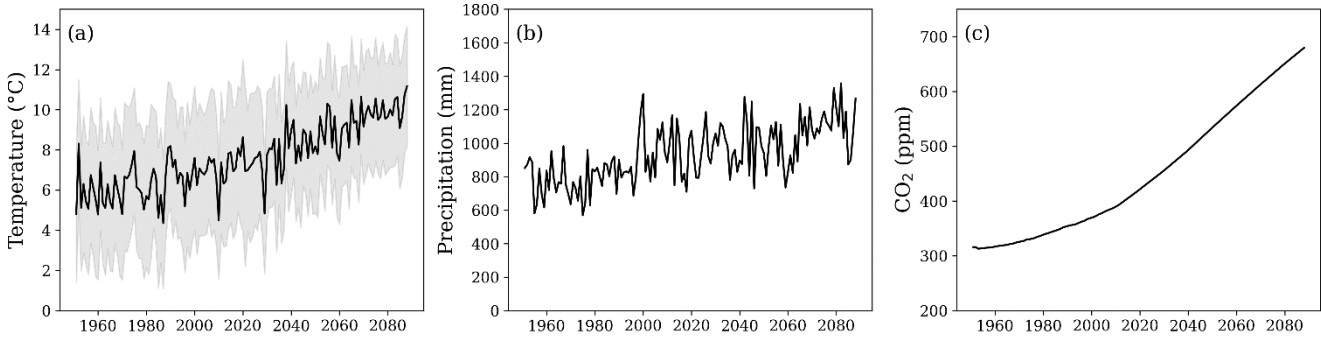

**Figure 2. (a) Annual mean daily air temperature (black line) and the range between the annual mean daily maximum and minimum air temperatures (grey area). (b) Annual precipitation. (c) Yearly average atmospheric $CO_2$ concentrations. The time series spans both the historical period (from 1951; data from the Swedish Meteorological and Hydrological Institute) and a future period (till 2088; data from model projections; see Section 2.2).**

The simulated forest stand is assumed to consist entirely of Norway spruce. The modelled forest management replicated historical events at the site: spruce planting in 1951, a 72% tree biomass thinning in 1979, a 10% biomass loss in 2010 due to storm damage, and a 96% biomass removal in 2019 as part of a clear-cutting operation. Harvesting plays a crucial role in regulating carbon dynamics in such systems. The large thinning event, which removed 72% of the biomass approximately 28 years after planting, represents a non-conventional management practice (Metzler et al., 2024). This intensive management strategy was incorporated into the simulations to accurately reflect the actual site's historical management. The second modelled rotation (2020–2088) followed the same biomass removal timing patterns as the first rotation.

The modelled soil profile, reflecting an average peat depth of 3 meters as reported by Nyström (2016) for the site, was discretised into 10 layers. Of these, the top nine layers were initially 0.2 m thick, while the bottom layer had a thickness of 1.2 m. At the onset of the simulation, all layers were characterised by the same properties. Bulk density was uniformly set to 0.20 $g_{soil}$ $cm^{-3}_{soil}$, informed by on-site observations and corroborated by findings in managed peat (Liu et al., 2020), while soil organic matter (SOM) content was set to 87% based on Meyer et al. (2013) and mineral soil content was set to 13%. Initial SOM was allocated entirely to the peat SOM compartment, and 50% of it was assumed to be soil organic carbon (SOC), which implied an initial soil carbon density of 0.08 $g_C$ $cm^{-3}_{soil}$. The initial carbon-to-nitrogen (C:N) ratio was set to 21, aligning with the observed average C:N at the site (Eriksson, 2021).

We set the initial ditch depth at 0.6 m based on ditch depth estimations from previous work conducted at the site (He et al., 2016; Nyström, 2016). We aimed to simulate standard ditch network maintenance (DNM) practices. In reality, the ditch was not maintained after clear-cutting in 2019 due to a rewetting experiment that began in 2022. Therefore, NEP observations for 2020 and 2021 were made after clear-cutting and during a period without DNM. To integrate historical accuracy with our aim of representing conventional management practices, we reset the ditch depth to 0.6 in our simulation starting in 2022. In the model formulation, lateral drainage is influenced by changes in ditch depth, which reflect variations in soil profile depth and hydraulic conductivity due to changes in the bulk density of the layers susceptible to lateral drainage. In reality, ditch depth is also influenced by infilling caused by sedimentation, vegetation growth, and bank erosion (Hökkä et al., 2020). However, these

processes are not incorporated into the model for the sake of simplicity. A more detailed description of the scenario parameterisation can be found in Supplementary Information 2.

## 2.3 Representativeness of model simulations

To evaluate the model's performance in replicating observed variables, we compared model outputs to available observations of abiotic factors controlling carbon dynamics and observations of carbon fluxes. For abiotic factors controlling carbon

dynamics, we focused on soil temperature and groundwater level (GWL), which are regarded as the main regulators of carbon fluxes in drained peatlands (Escobar et al., 2022; Evans et al., 2021; Jauhiainen et al., 2023). NEP data was available for the entire stand. In contrast, data for soil temperature and GWL from several locations at the site were averaged for the numerical comparison with the model estimates. We calculated the coefficient of determination ($R^2$) and the root mean squared error (RMSE) as goodness of fit measures.

For on-site observations, daily GWL data spanning six years (2014-2020) were available at four distinct locations. Concurrently, at three locations, daily soil temperature records covering 14 years (2008-2022) were obtained for three depths (0.05, 0.15, and 0.30 m). Measurement methods used at the site are described in Ernfors et al. (2011) and Klemedtsson et al. (2010). NEP (i.e. gross primary productivity minus ecosystem respiration) data were obtained from measurements done by eddy covariance. On-site NEP measurements were conducted in 2008 while trees were present, with subsequent data from

2020 and 2021 acquired post-clear-cutting, offering insights into soil respiration without substantial photosynthetic activity. NEP data processing and acquisition for the year 2008 is described in Meyer et al. (2013). For the years 2020-2021, the high-frequency data needed for flux calculations were acquired with an ultrasonic anemometer (USA-1, METEK GmbH, Germany) and a LI-7200RS gas analyser (LI-COR Biosciences, NE, USA) mounted at 2.15 m height above the low vegetation. The data acquisition frequency was 10 Hz, and the half-hourly average $CO_2$ flux was calculated with the EddyPro software, version

7.0.7 (LI-COR Biosciences, NE, USA) following the ICOS methodology (Sabbatini et al., 2018). Gaps in the dataset were subsequently filled using the REddyProCWeb online tool (Wutzler et al., 2018).

ForSAFE-Peat calibration was intentionally limited, as the objective was to evaluate the outcomes of common modelling assumptions under the site's specific conditions that inspired our simulation. We manually calibrated two parameters: the modifier of the bottom layer hydraulic conductivity that controls percolation ($limK_{sat}$) and the fraction of wood that respires

($RWF$). $limK_{sat}$ directly controls water leaving the soil profile by modulating percolation, thereby affecting the soil water balance. $RWF$ influences autotrophic respiration, which affects the tree's carbon balance and, consequently, biomass. In turn, biomass impacts water uptake, influencing groundwater levels. The water table also affects biomass by controlling water availability and nitrogen mineralisation. Calibration of these two parameters was conducted by comparing model outputs to GWL observations from two locations (2008–2013) and to biomass estimates for the site (2008–2010) derived from tree ring

data (He et al., 2016). Additional details are provided in Supplementary Information 2.

## 2.4 Carbon exchange metrics and system boundaries

Two metrics related to the carbon balance were selected to evaluate the potential effects of carbon exchanges on climate: the net carbon balance (NCB) and the integrated carbon storage (ICS). The NCB is calculated as:

$$NCB(T) = \int_{t_0}^{T}[Ic(t) - Oc(t)]\,dt, \tag{1}$$

where $NCB(T)$ is expressed in units of mass of carbon per ground area ($g_C$ $m^{-2}_{soil}$) and is calculated as the carbon gain or loss after integrating the input fluxes of carbon ($Ic(t)$) minus the outputs fluxes of carbon ($Oc(t)$) from the beginning of the period of analysis ($t_0$) until the end ($T$).

The $ICS(T)$ can be interpreted as the cumulative carbon storage and is calculated by integrating the $NCB(t)$ throughout the period of the analysis (Muñoz et al., 2024),

$$ICS(T) = \int_{t_0}^{T} NCB(t)\,dt. \tag{2}$$

Based on equation (2), this metric is expressed as the mass of carbon per ground area multiplied by time ($g_C$ yr $m^{-2}_{soil}$). The $ICS(T)$ is useful because it accounts for the time dynamics of carbon storage, which in turn control the cumulative contribution of a system to atmospheric cooling or warming (Muñoz et al., 2024; Sierra et al., 2021). When a system exhibits a very dynamic carbon exchange characterised by periods of large net losses and periods of large net gains, the $NCB(t)$ might vary between

positive and negative. The interval of time during which accumulated losses exceed accumulated gains can be interpreted as a period of negative effects on climate, while the opposite is true for the interval of time during which accumulated gains exceed accumulated losses. The cumulative effect of fluctuations in carbon storage is captured by the $ICS(T)$ via integration of $NCB(t)$ throughout the time period from $t_0$ to $T$. ICS has been proposed to account for carbon permanence in a system (Fearnside et al., 2000). Studies like Sierra et al. (2021) have shown that ICS can effectively account for the time carbon spends

stored in ecosystems, providing a more comprehensive means of analysing and comparing trajectories of carbon accumulation (Muñoz et al., 2024).

We estimated the previously explained metrics for three different system boundaries: the soil, the ecosystem and the ecosystem, including harvested wood products (ecosystem+HWP). Differences in system boundaries imply different inflows and outflows of carbon as quantified by equation (1). By examining different system boundaries, we can offer diverse perspectives on the

carbon exchanges (and thus the potential effect on climate) of drained forested peatlands. Additionally, these delineations provide valuable categories for analysing the temporal dynamics of carbon fluxes and their associated controlling factors. Differences in system boundaries are represented and explained in Figure 3.

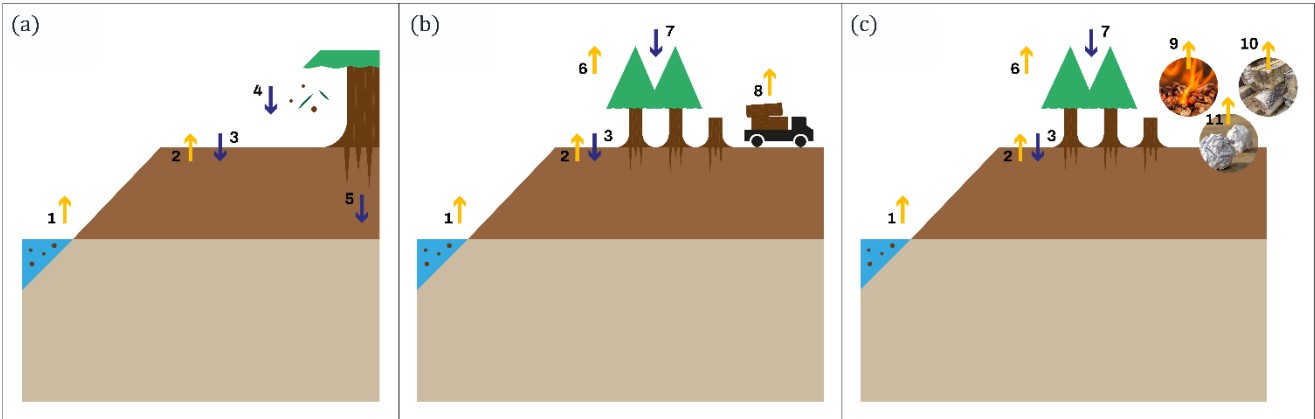

**Figure 3. System boundaries used in the study are (a) soil boundary, (b) ecosystem boundary, and (c) ecosystem + harvested wood products (HWP) boundary. Yellow arrows represent carbon outflows, and blue arrows represent carbon inflows: carbon leaching (arrow 1), soil-atmosphere carbon exchange (arrows 2 and 3), litterfall (arrow 4), belowground autotrophic respiration (arrow 5), aboveground autotrophic respiration (arrow 6), photosynthesis (arrow 7), harvested biomass (arrow 8), and outflows from the decay of HWPs (arrows 9, 10 and 11).**

For all system boundaries, outputs are represented by negative fluxes and inputs by positive fluxes. The soil boundary includes inflows from litterfall and belowground autotrophic respiration, with outflows from leached carbon (e.g., dissolved organic carbon, $CO_2$, and $CH_4$). Soil-atmosphere carbon exchange is gradient-controlled and can act as either an input or output of gaseous carbon ($CO_2$ and $CH_4$). At the ecosystem boundary, photosynthesis is an inflow, while leaching, aboveground autotrophic respiration and harvested biomass are outflows. Soil-atmosphere exchange is also included. The ecosystem +HWP boundary accounts for the same fluxes as the ecosystem boundary, but harvested biomass is replaced by the decay of wood products.

## 3 Results

### 3.1 Representativeness of model simulations

The model captured daily observations of groundwater table and soil temperature relatively well but less so for daily NEP. However, simulated annual and seasonal NEP are closely comparable to the observations.

### 3.1.1 Abiotic factors

Observed GWL from 2014 to 2021 had a mean of -0.40 m and a standard deviation of 0.17 m. Only considering the period before clear-cutting (2014-2019), observed GWL had a mean of -0.45 m. Summer lower values before clear-cutting ranged between -0.6 and -0.9 m. The high summer GWL observed after 2020 is attributed to the final felling of 2019, which decreased transpiration, thereby increasing GWL. Despite the considerable variance among observations at different locations, the simulations generally fell within the observed range and captured variations at both seasonal and dry-down time scales (Figure

4a). The R² between average observed and simulated water table depths was 0.78, and the RMSE was -0.08 m (b). Therefore, the model reliably reproduced observed GWL but with a clear, although relatively small, underestimation particularly apparent during winter. The model simulated lower GWL during winter compared to the average among the four locations. However, this lower water table is not expected to substantially impact soil $CO_2$ emissions, as low temperatures impede decomposition.

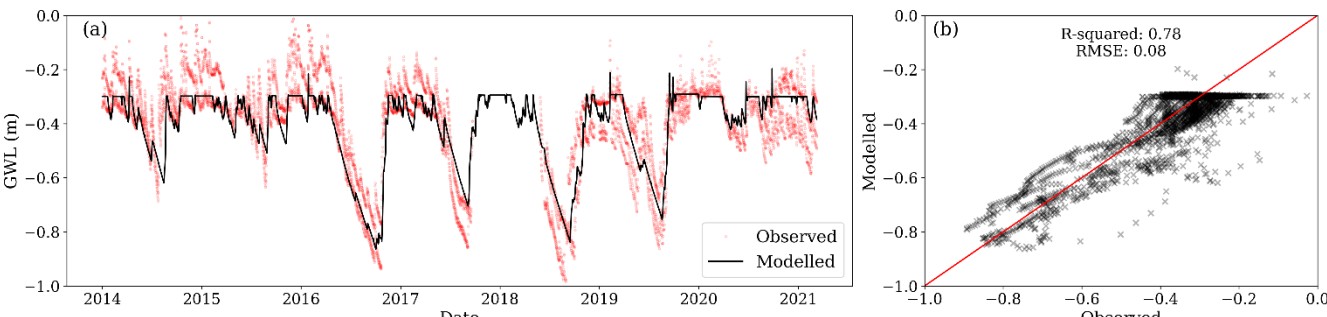

**Figure 4. (a) Modelled daily GWL (black line) and observations (red dots) from 4 different locations within the site; negative values mean distance to the surface. (b) Relationship between the mean observed daily GWL (averaged across locations) and modelled daily GWL.**

Daily soil temperature from 2008 to 2021 exhibited low variability between locations. The observed mean annual soil temperature at 0.05 m depth was 5.7 °C, and the standard deviation was 5.4 °C. Simulated soil temperature in the first layer correlated strongly with the average observed temperature at 0.05 m among the three measurement locations (R² of 0.77, RSME of 2.57 °C), as shown in Figure 5b. Similar comparisons of soil temperature at depths of 0.15 and 0.30 m are given in Appendix C, with $R^2$ values $\geq$ 0.76. Simulated soil temperature showed slight but consistent overestimations compared to observations during spring and summer, which could lead to an overestimation of the decomposition temperature modifier function (Figure 5a).

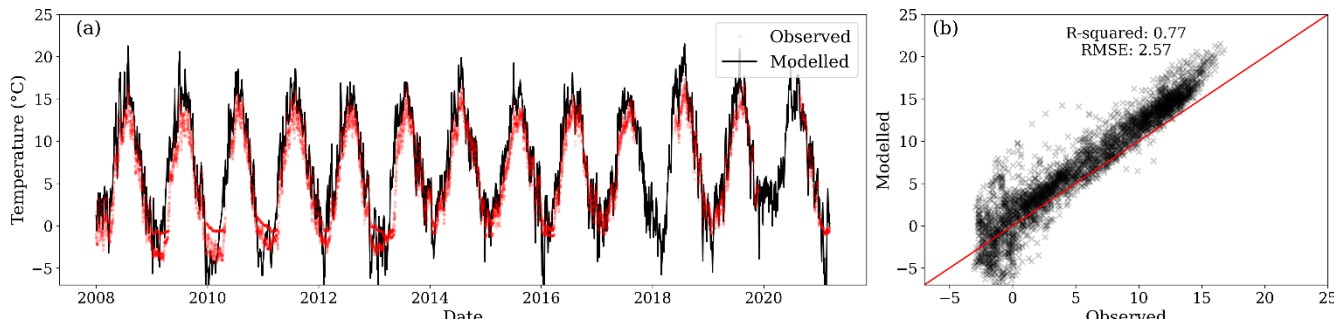

**Figure 5. (a) Modelled daily soil temperature for the first layer (black line) and observations at 0.05m depth (red dots) from 3 locations within the site. (b) Relationship between observed and modelled soil temperature values. During the comparison period, the first layer's centroid was between 0.077 m and 0.081 m.**

## 3.1.2 Carbon fluxes

NEP measurements revealed that the site acted as a net sink of $CO_2$ in 2008 while still forested, transitioning to a $CO_2$ source in 2020 and 2021 after clear-cutting. While during 2008, the mean NEP was 0.55 $g_C$ m$^{-2}$soil d$^{-1}$; during 2020 and 2021, the mean NEP was -1.08 and -0.59 $g_C$ m$^{-2}$soil d$^{-1}$, respectively. Despite reproducing soil temperature and GWL reasonably well on a daily basis, the model failed to capture daily changes in NEP (Figure 6a). However, when aggregated to seasonal values, the model performed adequately. For fluxes aggregated over warm months (May, June, July, August, September and October) and cold months (November, December, January, February, March, and April), the model achieved $R^2$ = 0.94 and RMSE = 40.8 $g_C$ m$^{-2}_{soil}$ half-yr$^{-1}$ (Figure 6b). The model successfully captured the site transition from a carbon sink to a source. Observed annual NEP fluxes for 2008, 2020, and 2021 were 204, -396, and -216 $g_C$ m$^{-2}_{soil}$ yr$^{-1}$, respectively, while the model estimated values of 258, -282, and -270 $g_C$ m$^{-2}$soil yr$^{-1}$, respectively.

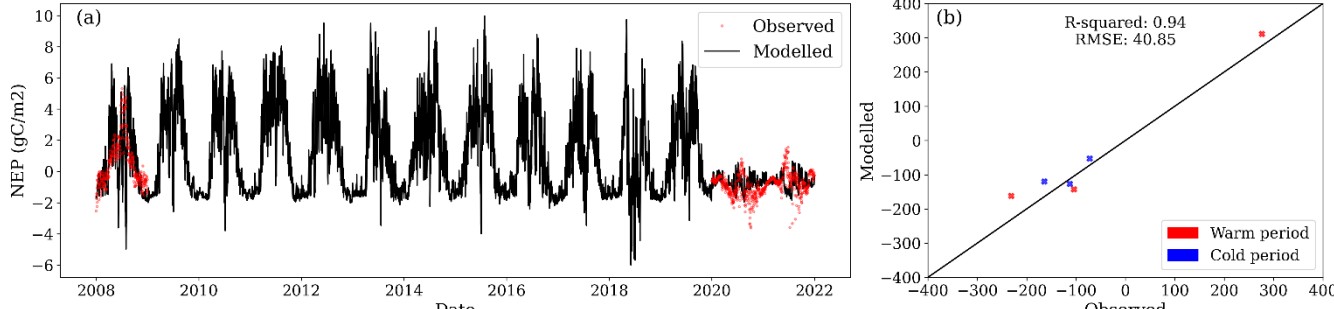

**Figure 6. (a) Modelled daily net ecosystem productivity (black line) and observations (red dots). (b) Relationship between observed and modelled net ecosystem productivity values. Values for model evaluation correspond to the aggregation of fluxes into warm (May, June, July, August, September and October) and cold (November, December, January, February, March, and April) months of the year.**

## 3.2 Carbon exchange dynamics across system boundaries.

The simulated NCB and ICS were negative under all system boundaries at the end of the second rotation, with the exception of the NBC at the ecosystem+HWP scale. Both metrics showed strong and similar sensitivity to system boundaries (Table 1). Expanding the system boundaries positively influenced both the NCB and ICS, with soil showing the most negative values, followed by the ecosystem and then the ecosystem+HWP. Under the ecosystem+HWP boundary, although the system accumulated more carbon than it lost by the end of the simulation, the ICS remained negative, indicating a potential persistent negative effect on climate over the same period.

**Table 1. Net carbon balance (NCB) and integrated carbon storage (ICS) for the soil, ecosystem and ecosystem+HWP as system boundaries at the end of two forest rotations. HWP refers to harvested wood products.**

| System boundaries | NCB ($g_C$ m$^{-2}$$_{soil}$) | ICS ($g_C$ yr m$^{-2}$$_{soil}$) |
|---|---|---|
| **Soil** | -34897 | $-2.42 \times 10^6$ |
| **Ecosystem** | -25249 | $-1.20 \times 10^6$ |
| **Ecosystem+HWP** | 2307 | $-0.59 \times 10^6$ |

The main carbon dynamics during the simulation are depicted in Figure 7. Within the soil, the peat stock decreased with time. Peat losses were not compensated by soil stocks associated with litter and biomass residues despite large increments in those stocks during the second rotation. The plant carbon stocks were modulated by the cycle of forest management and environmental conditions, which increased plant carbon during the second rotation. HWP carbon stocks substantially increased after 2019's clear-cutting.

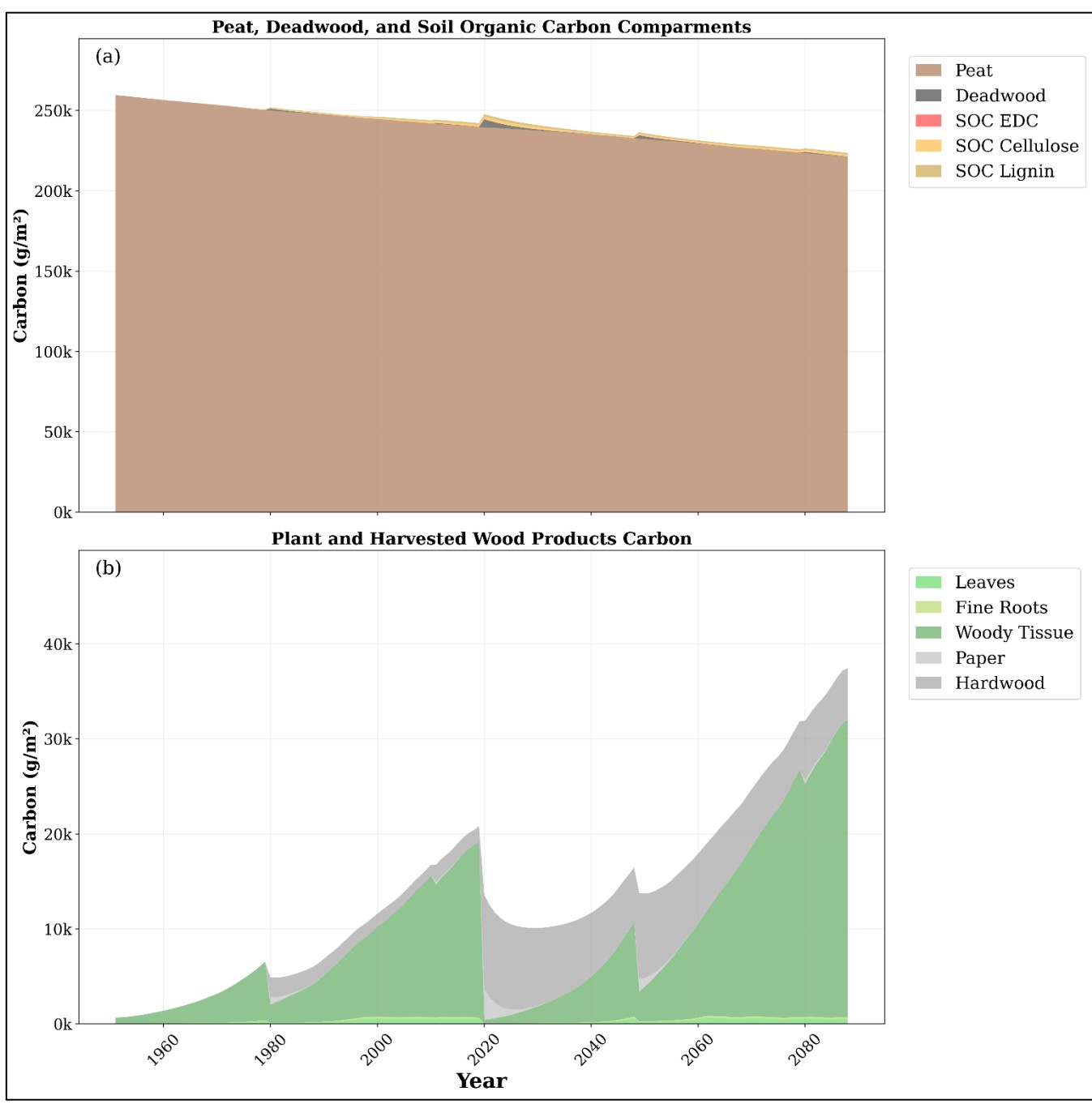

**Figure 7. Temporal evolution of main carbon stocks during the simulation. a) Carbon stocks in peat, deadwood, easily decomposed compounds (EDC), cellulose and lignin. b) Carbon stocks in leaves, fine roots, woody tissue (stem plus branches), and paper from harvested wood and hardwood products. Note the difference in scale of the y-axis between the upper and lower plots.**

### 3.2.1 Soil carbon dynamics

At the end of the second rotation, for the soil system alone, the NCB was -34897 $g_C$ m$^{-2}$$_{soil}$, while the ICS was -2.42×10$^6$ $g_C$ yr m$^{-2}$$_{soil}$. The NCB declined consistently over time, with the exception of transient recovery events associated with inputs of harvest residues (Figure 8b). This reflects the persistent net loss of carbon despite the continuous inputs of litter to the soil (Figure 8a). The ICS declined exponentially with time, as it accounts for the compounding effects of the emitted carbon residing in the atmosphere instead of in the soil or vegetation (Figure 8c).

The average annual carbon balance within the soil amounted to -252.8 $g_C$ m$^{-2}$$_{soil}$ yr$^{-1}$, showing virtually no differences between the first and second rotations. Key inflows included litterfall and carbon transfers from deadwood, primarily dead stumps left after harvest. The site functioned as a small $CH_4$ sink, except during harvesting years when it became a slight $CH_4$ source. The most substantial outflow was through soil $CO_2$ emissions, whereas leached carbon (DOC, $CH_4$ in water and $CO_2$ in water) contributed only 15% of total outflows.

The annual balance was lowest at the onset of the forest rotation due to low litter input and substantial soil $CO_2$ emissions from peat decomposition. For example, the annual soil balance was -330 $g_C$ m$^{-2}$$_{soil}$ yr$^{-1}$ during the first five years of the first forest rotation, while it was -223 $g_C$ m$^{-2}$$_{soil}$ yr$^{-1}$ during the last 8 years. As the tree stand matured, the balance became less negative, occasionally turning positive during years with large litterfall inputs (e.g., 908 $g_C$ m$^{-2}$$_{soil}$ yr$^{-1}$ as a result of clear-cutting at the end of 2019).

Litter inputs increased during the second rotation (144 $g_C$ m$^{-2}$$_{soil}$ yr$^{-1}$) compared to the first rotation (118 $g_C$ m$^{-2}$$_{soil}$ yr$^{-1}$), thanks to larger tree biomass. Litterfall increased as the forest stand aged due to its relation with biomass size. Litterfall from leaves and roots represented 81 % of the total litterfall, the rest being associated with branches and bark. Deadwood carbon transfer became particularly noteworthy in 2020 following clear-cutting, compensating for low litter from small trees at the outset of the second rotation, becoming the primary input for the first 18 years of this rotation. During the first rotation, deadwood

transfer from dead stumps left after removals from management was, on average, 43 $g_C$ m$^{-2}$$_{soil}$ yr$^{-1}$, while during the second rotation, it was substantially higher, amounting to 123 $g_C$ m$^{-2}$$_{soil}$ yr$^{-1}$.

$CO_2$ emissions from the soil were also higher during the second rotation (-506 $g_C$ m$^{-2}$$_{soil}$ yr$^{-1}$) compared to the first rotation (-380 $g_C$ m$^{-2}$$_{soil}$ yr$^{-1}$), partly due to increased carbon inputs from litter and deadwood, resulting in higher $CO_2$ emissions from the easily decomposed compounds (EDC), cellulose and lignin SOM compartments. Nonetheless, emissions from peat

decomposition remained the primary source of $CO_2$ throughout the simulation, with similar magnitudes between rotations. The decreasing availability of peat in the first three soil layers, resulting from reduced peat mass due to decomposition, did not lead to lower decomposition fluxes because increasing soil temperature promoted decomposition. During the first rotation, the average peat decomposition rate constants for the first, second and third soil layers were 0.012, 0.011 and 0.007 yr$^{-1}$, respectively; for the second rotation, the rate constants were 0.014, 0.013 and 0.011 yr$^{-1}$. Additionally, peat available for aerobic

decomposition from layers affected by ditch maintenance after 2022 further supported persistent and high peat decomposition rates.

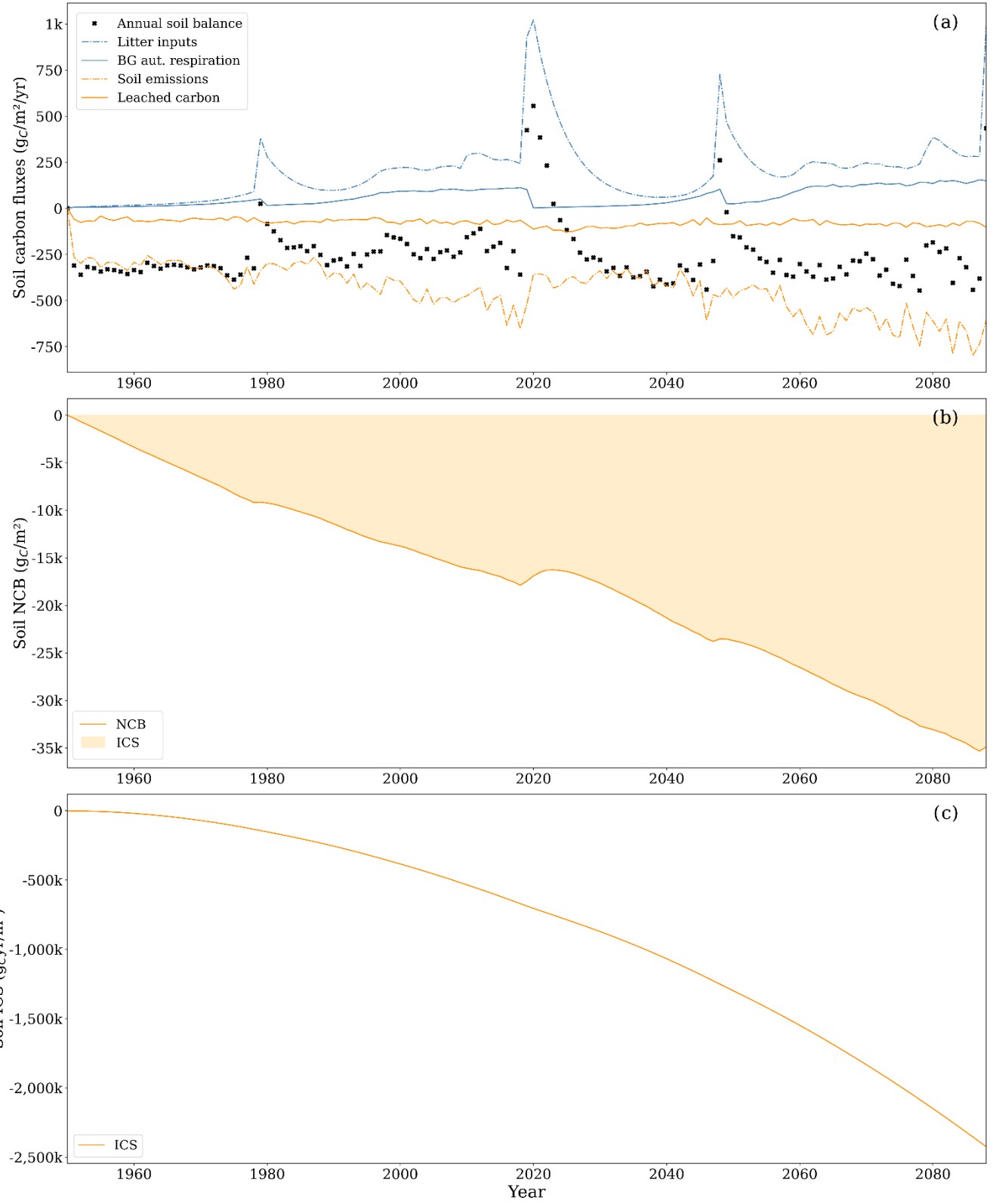

**Figure 8. (a) Litterfall and dead wood transfers to soil (dot-dashed blue line), autotrophic belowground respiration (solid blue line), $CO_2$ emissions from the soil (orange dot-dashed line), leached carbon composed of $CO_2$ and DOC (orange solid line) and soil net yearly balance (black cross). Note that, for the soil system, autotrophic belowground respiration is an inflow of carbon to the soil. Uptake of $CH_4$ and leached $CH_4$ were excluded from the graph as they comprised less than 1% of the total carbon flux. (b) net carbon balance (solid line) during the analysis period and the integrated carbon storage (shaded area). (c) integrated carbon storage (solid line).**

### 3.2.2 Ecosystem carbon dynamics

Focusing solely on the soil boundaries overlooks the primary mechanism through which drained forested peatlands accumulate carbon, which is the living tissue of trees. Therefore, analysing carbon dynamics within the ecosystem boundaries becomes essential. Under these boundaries, both metrics reveal a system with a less negative carbon balance compared to the soil system but still negative throughout the analysis period (Figure 9). By the end of the second rotation, NCB was -25249 $g_C$ m$^{-2}_{soil}$, while ICS was -1.20×10$^6$ $g_C$ yr m$^{-2}_{soil}$. Both metrics became more negative from the end of the first rotation to the end of the second rotation.

Under these boundaries, the average annual carbon balance amounted to -182 $g_C$ m$^{-2}_{soil}$ yr$^{-1}$. The balance turns positive if harvest years are not accounted for (136 $g_C$ m$^{-2}_{soil}$ yr$^{-1}$). Inflows were primarily driven by the spruce GPP (i.e. the site was a small net sink of $CH_4$). At the same time, the most important outflows included aboveground autotrophic respiration, $CO_2$ emissions from the soil, and biomass harvesting, accounting for 35%, 34%, and 25% of total outflow, respectively. Aboveground respiration increased throughout the rotation because it is primarily controlled by plant biomass. On average, aboveground respiration accounted for 40% of GPP, with lower values during the initial years of the simulation (around 34% in the first nine years of the first rotation) and higher values as aboveground woody biomass became a higher proportion of the plant biomass (around 46% in the last nine years of the first rotation). Notably, changes between rotations were minimal.

Soil $CO_2$ emissions followed a different trajectory. In the initial 9-year period, they represented 226% of GPP. However, as GPP increased faster than soil emissions, their relative contribution declined to 30% in the last 9 years of the first rotation. Similar values were observed in the subsequent rotation.

The temporal dynamics of flows explain NCB and ICS time trajectories. The rapid increase of GPP gradually offset early carbon losses during the rotation. The tree biomass stores a fraction of the GPP, partially offsetting the accumulated soil losses until harvest removes tree biomass, reducing the accumulated balance again. The metrics became more negative during the second rotation because sustained soil carbon losses were compounded with the soil carbon losses of the first rotation that were not compensated at the end of the first rotation.

During the second rotation, GPP notably increased to 1379 $g_C$ m$^{-2}_{soil}$ yr$^{-1}$, compared to 889 $g_C$ m$^{-2}_{soil}$ yr$^{-1}$ during the first rotation, so that the tree biomass at the end of the second rotation was 59% higher than in the first rotation. The increased photosynthetic rates are primarily attributed to the positive effect of higher atmospheric $CO_2$ concentration and higher temperature embedded in the model formulation. This sets off a reinforcing loop where higher potential photosynthesis leads to increased biomass growth, resulting in a higher leaf area index (LAI), further boosting photosynthesis. However, this process can be counterbalanced by several factors, including self-shading, foliar nitrogen dilution and water limitation. While in both

rotations, maximum LAI values were similar around 6.2 $m^2_{leaf}$ $m^{-2}_{soil}$, the average LAI during the first rotation (2.7 $m^2_{leaf}$ $m^{-2}_{soil}$) was lower than the average value for the second rotation (3.2 $m^2_{leaf}$ $m^{-2}_{soil}$), indicating that trees achieved maximum canopy

faster in the second rotation. The average foliar nitrogen content expressed as a percentage of leaf dry weight remained similar between rotations (1.52%). This was supported by consistently high nitrogen mineralisation. During the first rotation, the average yearly nitrogen mineralisation was 7.02 $g_N$ m-2soil yr-1, while during the second rotation, the average yearly nitrogen mineralisation was 10.63 $g_N$ $m^{-2}_{soil}$ $yr^{-1}$. Similarly, water limitation was unimportant in either rotation, with the ratio between actual plant water uptake and potential plant water uptake remaining at 0.98 for both. However, in some dry years, such as

2018, the ratio decreased to 0.91.

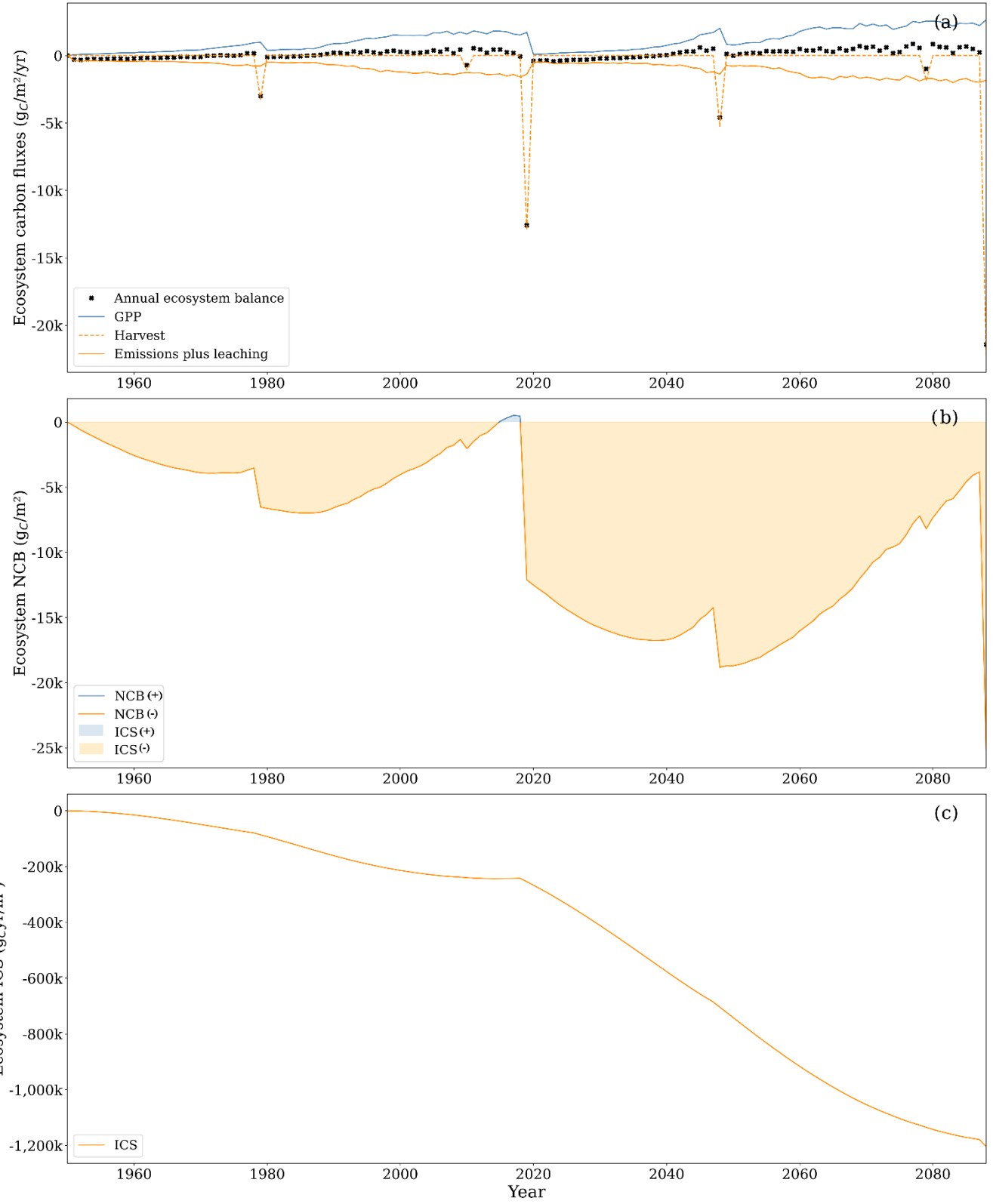

**Figure 9. (a) Gross primary productivity (solid blue line), aboveground respiration, soil CO₂ emissions, and leached carbon composed of CO₂ and DOC (orange solid line), carbon outputs due to harvesting (orange dashed line) and ecosystem net yearly balance (black cross). Uptake of CH₄ and leached CH₄ were excluded from the graph as they comprised less than 1% of the total carbon flux. (b) net carbon balance (solid line) and integrated carbon storage (shaded area). (c) integrated carbon storage (solid line)**

In years without harvest or extreme climatic conditions, GPP typically remains sufficiently high to offset ecosystem carbon losses, except during the initial rotation years when LAI is less than 1.5 $m^2_{leaf}$ $m^{-2}_{soil}$. Extreme climatic conditions can lead to a negative annual carbon balance, even in mature stands with high photosynthetic capacity (i.e. LAI > 5 $m^2_{leaf}$ $m^{-2}_{soil}$). For instance, in 2018, when precipitation was 25% below the average for 2005-2019, the annual ecosystem balance was -56 $g_C$ $m^{-2}_{soil}$ $yr^{-1}$. Conversely, in 2015, with precipitation 8% above the average, the balance was 451 $g_C$ $m^{-2}_{soil}$ $yr^{-1}$. During 2018, GPP was 84% of 2015 GPP due to water limitations during the summer. Aboveground respiration remained high in 2018 despite lower growth due to plant maintenance respiration, suggesting that the size of the forest stand and temperature can amplify the negative effect on carbon fluxes of dry years. Soil emissions in 2018 were 137% of the 2015 emissions, suggesting that water limitation to decomposition in the upper soil layers was overridden by aerobic decomposition in deeper peat layers that remained moist (Figure 10).

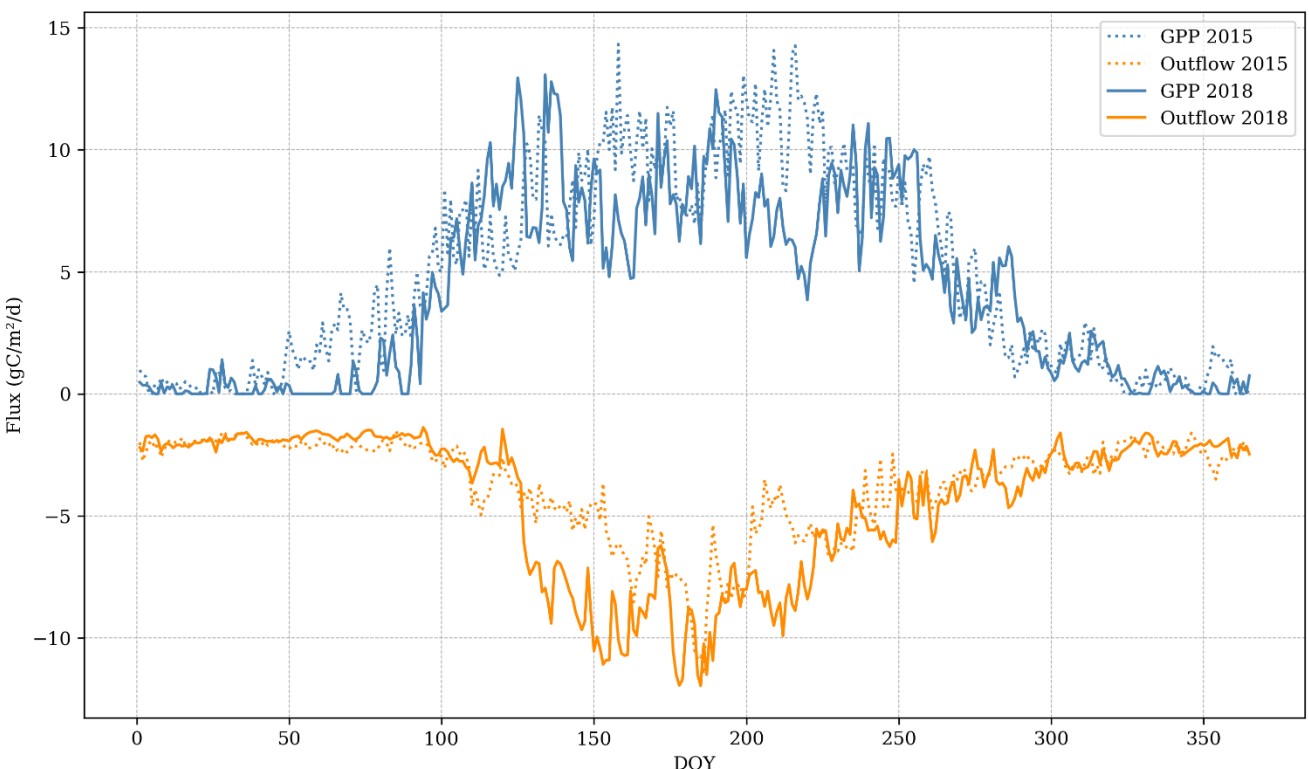

**Figure 10. Comparison between main ecosystem fluxes between a dry year (2018) and a typical year (2015). Outflow is comprised of soil CO₂ emissions, leached carbon and aboveground respiration.**

During most years, carbon accumulation in the plant compartment is more than the carbon lost by the soil compartments. Therefore, overlooking the removal of plant carbon by harvesting—amounting to 25% of the total carbon outflow from the ecosystem—could falsely suggest a carbon sink within the system. Across the two rotations, harvesting contributed to a total outflow of -46534 $g_C$ $m^{-2}_{soil}$ out of the 156550 $g_C$ $m^{-2}_{soil}$ photosynthesised by the plants.

### 3.2.3 Ecosystem+HWP carbon dynamics

Under the ecosystem boundaries, carbon associated with harvested biomass is treated as an outflow, as if harvested carbon were in the form of $CO_2$ or DOC. However, harvested wood does not undergo rapid conversion to $CO_2$. Consequently, the ideal boundaries for assessing effects on climate are those in which all outflows from the system ultimately leave as $CO_2$. Within the ecosystem+HWP system boundaries, harvested wood fate is tracked until its degradation into $CO_2$, providing a long-term perspective on potential effects on climate. By the end of the second rotation, the NCB turned positive at 2307 $g_C$ $m^{-2}_{soil}$, while ICS was large and negative at $-0.59 \times 10^6$ $g_C$ yr $m^{-2}_{soil}$ (Figure 11). Both metrics declined by the end of the second rotation compared to the end of the first rotation, when NCB was 2380 $g_C$ $m^{-2}_{soil}$, and ICS was $-0.17 \times 10^6$ $g_C$ yr $m^{-2}_{soil}$. This trend can be attributed to the compensation of early soil carbon losses by forest growth that is not completely cancelled by harvesting due to the slow decay of some harvested carbon. The capacity of wood products to hold carbon for some time resulted in a positive carbon balance by the end of each rotation. However, the ICS displays a negative trend because the initial losses are substantial, and later compensation is neither sufficient nor sustained long enough to counterbalance the extent and duration of the negative carbon balance under these system boundaries.

In the ecosystem+HWP system boundaries, the primary inflow of carbon was GPP, with negligible soil uptake of $CH_4$. The most important outflows were, in order of importance, aboveground respiration, soil carbon emissions, decay of harvested wood products and soil carbon leaching.

The decay of harvested wood products peaks in the year of harvest, directly proportional to the harvested biomass, with temporal dynamics independent of GPP fluctuations. During the second rotation, carbon loss from decaying harvested wood products accounted for 8% of total GPP. These outflows increased in the second rotation due to the 2019 clear-cutting, which transferred a substantial amount of carbon to harvested wood products. In the first rotation, carbon outflows from decaying harvested wood products were approximately 10% of aboveground respiration and soil $CO_2$ emissions, while in the second rotation, this proportion increased to 40%. The total outflow from decaying harvested wood products in the second rotation was 16,269 $g_C$ $m^{-2}_{soil}$, with half occurring between 2020 and 2043.

Although accounting for the slow decay of harvested wood products moderates the impact of clear-cutting on the net carbon balance, this intense harvesting process still caused a drastic shift at the end of the first rotation. From 1987 until 2019, the annual rate of change of the NCB was positive (233 $g_C$ $m^{-2}_{soil}$ yr$^{-1}$), which led to a positive NCB from 2010 until 2019. During that period, the ICS negative trend slowed down. Clear-cutting in 2019 quickly reduced the NCB, which experienced a strong negative rate of change during the first 10 years of the second forest rotation (-960 $g_C$ $m^{-2}_{soil}$ yr$^{-1}$). The NCB was projected to become slightly positive only in 2083. The year after clear-cutting, GPP was reduced by 95% compared to the previous year,

while soil emissions remained high. The decay of harvested wood products exceeded GPP for the first eight years of the second

rotation.

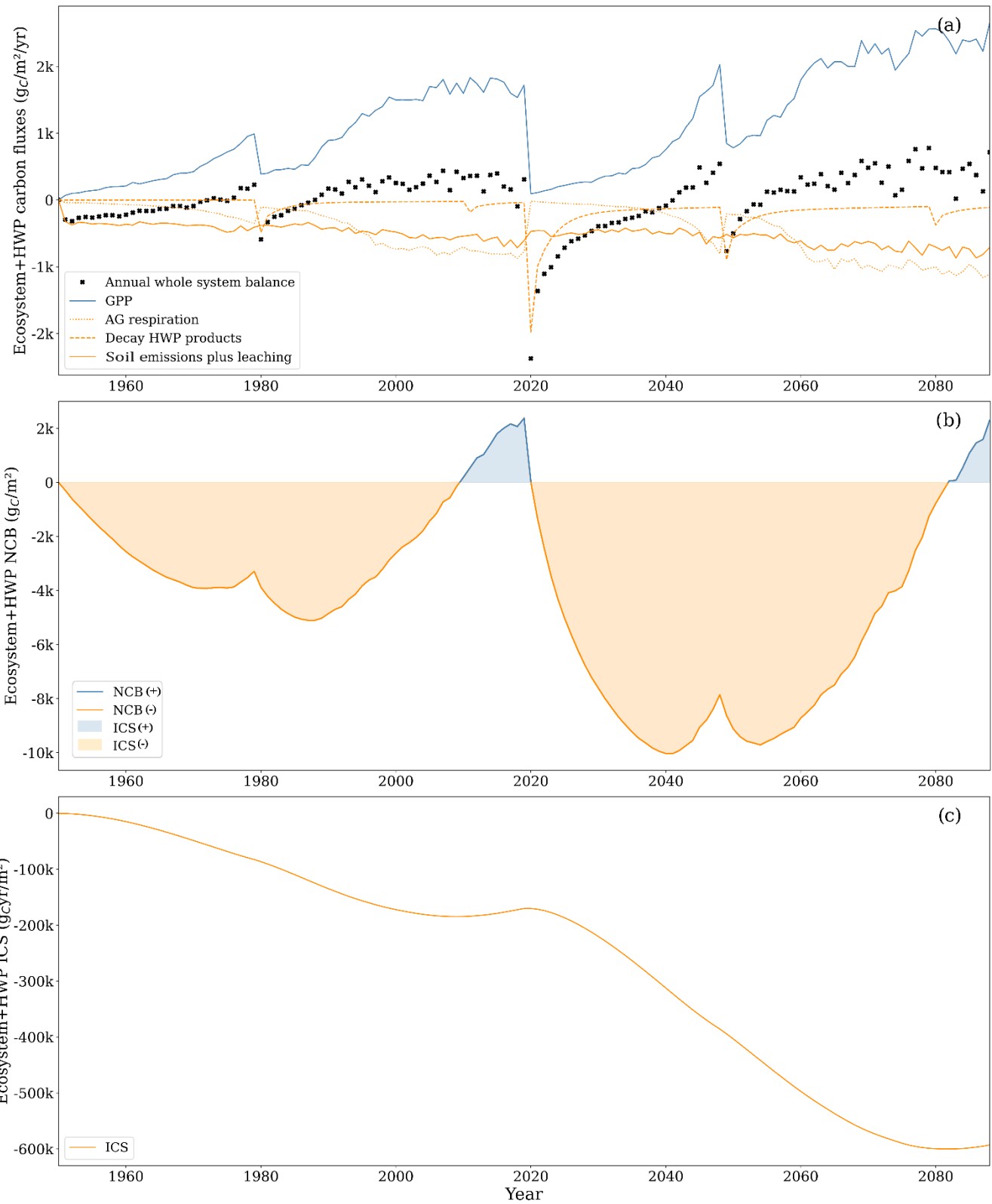

**Figure 11. (a) Gross primary productivity (solid blue line), aboveground respiration (orange dotted line), soil carbon losses comprising soil CO₂ emissions and leached carbon composed of CO₂ and DOC (orange solid line), CO₂ from decay of harvested wood products (orange dashed line) and ecosystem + harvested wood products net yearly balance (black cross). Uptake of CH₄ and leached CH₄ were excluded from the graph as they comprised less than 1% of the total carbon flux. (b) net carbon balance (solid line) and integrated carbon storage (shaded area). (c) integrated carbon storage (solid line).**

## 4 Discussion

### 4.1 On the representativeness of simulated carbon dynamics and the abiotic context

Site conditions can substantially influence the magnitude of carbon fluxes in northern drained forested peatlands. Therefore, soil emission factors for this land category are classified based on nutrient availability, climate conditions, and drainage level (Jauhiainen et al., 2023; Wilson et al., 2016).

In our simulated peatland, nutrient conditions are primarily determined by an initial soil organic matter C:N ratio of 21. Under these conditions, sites are often classified as *Herb-rich* type (Ojanen et al., 2010) or eutrophic (Minkkinen et al., 2020). The average modelled soil temperature of 7.0°C from 1990 to 2020 aligns with values observed in cool temperate or hemiboreal sites in southern Sweden and Estonia (Minkkinen et al., 2007; Ranniku et al., 2024).

Regarding drainage, during the first rotation, the mean annual GWL was -0.43 m—comparable to other well-drained forested peatlands (Leppä et al., 2020; Maljanen et al., 2012; Menberu et al., 2016). Slightly lower values were observed toward the end of the rotation due to peat subsidence. Reported mean annual GWL values typically range between -0.3 m and -0.5 m at distances of 5 m and 15 m from the ditch, respectively (Haapalehto et al., 2014). Water table fluctuations simulated by ForSAFE-Peat reflect those of a well-drained site with functional ditches.

Based on these characteristics, our simulated site can be classified as nutrient-rich and well-drained, with a climate that falls between boreal and temperate regions. This classification helps explain why our estimates of soil carbon balance during the first rotation (-252 $g_C$ $m^{-2}$ $yr^{-1}$) are similar to those found in drained forested peatlands previously used for agriculture in cool temperate regions (-256 $g_C$ $m^{-2}$ $yr^{-1}$) by a metanalysis of field-based observations (Jauhiainen et al., 2023). These values are at the higher end of estimations for the more general category of drained forested peatlands in northern latitudes, but they are still well within the variability reported (Jauhiainen et al., 2023; Jovani-Sancho et al., 2021).

It is important to note that our study site was actively afforested, as it was initially an open fen. While peatlands with substantial tree cover are relatively rare in the UK, spruce and pine mires are more common in Sweden and even more prevalent in Finland (Laine et al., 2006). Peatlands without prior tree cover before drainage are likely to have higher soil water saturation levels than those with substantial tree cover (Beaulne et al., 2021). On peatlands with substantial tree cover, carbon accumulation in the upper soil layers is likely more dependent on stabilisation mechanisms occurring under aerobic conditions (Kilpeläinen et al., 2023). In contrast, carbon in peatlands without substantial tree cover is more likely stabilised by anoxic conditions, making it more sensitive to water table drawdown. Such nuances in carbon dynamics are often lost in the broad categories used to account for carbon in these land-use systems.

Comparable observations for soil $CO_2$ emissions to our simulations are mostly limited to dark chamber measurements that did not remove litter and include belowground autotrophic respiration. For example, Arnold et al. (2005) estimated soil $CO_2$ emissions at -392 $g_C$ $m^{-2}$ $yr^{-1}$ for a drained forested peatland with a 50-year-old Norway spruce stand. In contrast, our estimation for the first rotation, when our spruce stand was between 45 and 55 years old, was -442 $g_C$ $m^{-2}$ $yr^{-1}$. While these sites shared some similarities, such as the grown tree species, the site described by Arnold et al. (2005) had lower nitrogen availability

(C:N ratio of 28), lower carbon content (soil carbon density of 0.07 $g_C$ $cm^{-3}_{soil}$), and a higher water table (-0.27 m). Overall, data in the literature exhibits large variability, and our estimations tend to fall on the higher end of this range.

DOC leaching is often assumed to be a less important component of the soil carbon outflux than soil $CO_2$ emissions and is not often reported. Wilson et al. (2016) estimated -30 $g_C$ $m^{-2}$ $yr^{-1}$ for temperate drained forested peatlands, but the lack of data did not allow for separate fluxes by nutrient status. We calculated an average of -34 $g_C$ $m^{-2}$ $yr^{-1}$ during the first rotation. Our ratio

of soil $CO_2$ emissions to DOC exports of 0.09 during the first rotation was similar to the 0.12 of Wilson et al. (2016). Interestingly, the ratio between leached DOC and GPP in our study (0.08) was around the higher end (range: 0.002 to 0.08) estimated for Swedish watersheds by Manzoni et al. (2018). Notably, these ratios in our study were much higher during the first years of rotation due to the effect of drainage and the absence of substantial photosynthetic activity, highlighting the impacts of processes such as clear-cutting (Gundale et al., 2024).

Soil carbon losses, in the form of $CO_2$ emissions and DOC leaching, can be offset by litter inputs. Litterfall rates in Norway spruce exhibit considerable variability and are highly sensitive to nutrient status (Kleja et al., 2008). In nutrient-rich conditions, measured litterfall rates for Norway spruce have been estimated at 150 and 301 $g_C$ $m^{-2}$ $yr^{-1}$ (Blaško et al., 2022), which are similar to our simulated average of 198 $g_C$ $m^{-2}$ $yr^{-1}$ when the LAI exceeded 2.0 $m^2_{leaf}$ $m^{-2}_{soil}$. Our average litter production rate for the first rotation (118 $g_C$ $m^{-2}$ $yr^{-1}$) was lower than the 219 $g_C$ $m^{-2}$ $yr^{-1}$ reported in a modelling study by Kleja et al. (2008),

which simulated nutrient-rich conditions at a site in southern Sweden using the COUP model. However, values measured in a Norway spruce stand in Sweden (Hansson et al., 2013). Finland (Hilli, 2013), Estonia (Uri et al., 2017), and Latvia (Bārdule et al., 2021) are comparable to our estimates.

The magnitude of litterfall is closely tied to plant biomass, which increases with GPP fluxes (Ojanen et al., 2014). Using partitioning assumptions, GPP observations are often derived from net ecosystem production (NEP) measurements. For

instance, Mamkin et al. (2023) reported a five-year average GPP of 1494 $g_C$ $m^{-2}$ $yr^{-1}$ for old Norway spruce on peat in Russia with an average LAI of 3.5 $m^2_{leaf}$ $m^{-2}_{soil}$, comparable to our estimation of 1295 $g_C$ $m^{-2}$ $yr^{-1}$ for a similar LAI value (3.56 $m^2_{leaf}$ $m^{-2}_{soil}$) during the first rotation. Additionally, Korkiakoski et al. (2023) estimated a five-year average GPP of 1406 $g_C$ $m^{-2}$ $yr^{-1}$ in a nutrient-rich drained forested peatland with spruce and pine in the south of Finland, with an LAI slightly above 2 $m^2_{leaf}$ $m^{-2}_{soil}$ determined using remote sensing. Our GPP estimations for similar LAI values were around 1128 $g_C$ $m^{-2}$ $yr^{-1}$. However,

simulated GPP values during the first two years after clear-cutting (93 and 110 $g_C$ $m^{-2}$ $yr^{-1}$) were lower than those reported by Korkiakoski et al. (2019) for a nutrient-rich drained peatland following clear-cutting (179 and 301 $g_C$ $m^{-2}$ $yr^{-1}$). This discrepancy highlights the importance of non-tree vegetation in sustaining GPP after clear-cutting, which was captured by Korkiakoski et al. (2019) but not accounted for in our model.

GPP in the second rotation increased by 64%, driven by higher temperatures and elevated atmospheric $CO_2$ concentrations

under a context of nitrogen and water availability. Increased photosynthetic activity in Norway spruce under free air $CO_2$ enrichment experiment has been documented. (Bader et al., 2016) reported an increase of 73% in the photosynthetic rate of the upper-canopy shoots with an atmospheric $CO_2$ concentration increase of 150 ppm. Similarly, Sigurdsson et al. (2002) observed a 53% increase in the rate of light-saturated photosynthesis under high nitrogen availability with an atmospheric $CO_2$ concentration increase of 350 ppm.

Under a reasonable abiotic regime, defined by realistic water table depth and soil temperature, the decomposition representation used in ForSAFE-Peat—similar to those of models like ORCHIDEE and LPJ—produces credible estimates of peat losses. Additionally, the PnET default parameterisation for carbon assimilation, respiration, and litterfall, combined with the calibrated respiring wood fraction, yields realistic tree biomass accumulation. These results align closely with values reported in the literature for similar systems, supporting the model's ability to simulate carbon dynamics in drained forested

peatlands under comparable conditions.

## 4.2 Model limitations

ForSAFE-peat reproduced GWL and soil temperature observations with reasonable accuracy. However, it simulates lower GWL during winter, which could be related to its omission of freezing effects on water flow or excessively fast lateral water flow associated with drainage. The simple approach used to simulate drainage may also fail to capture critical hydrological

dynamics, such as anisotropic hydraulic conductivity, ditch geometry, or water-induced soil volume changes through peat swelling.

Regarding temperature, the model tends to overestimate spring and summer temperatures. This overestimation may result from a lower GWL at the onset of spring compared to observations, which reduces heat capacity and possibly heat conductivity. Furthermore, the discrepancy might arise from the model not accounting for the temperature modulation by trees, mainly

through evapotranspiration latent heat fluxes and canopy shading.

Despite these shortcomings, the model provides a reasonable abiotic context for assessing carbon dynamics. It is essential, however, to evaluate the limitations of the representation of the carbon cycle and its implications in our results. The model followed commonly used formulations for peat soils (Kleinen et al., 2012; Qiu et al., 2018), where peat is defined as a conceptual compartment with unspecified chemistry that decomposes following first-order exponential decay, with rate

constants modified by environmental conditions. Even though this description provides reasonable carbon dynamics at yearly and decadal time scales, limitations within this representation might explain why the model did not represent daily NEP fluxes well. Firstly, in the current model structure, decomposition rates increase slowly with moisture content until field capacity, but this response could be faster in peat soils (Rewcastle et al., 2020; Ťupek et al., 2023). Furthermore, the complex redox chain that controls decomposition in peat soils is simplified by a function only considering water content. In reality, even unsaturated

conditions can lead to anoxia if intense decomposition depletes oxygen (Fan et al., 2014). Conversely, fully saturated

conditions might not form methane if electron acceptors like nitrate or sulphate are available (Cui et al., 2024; Reddy and DeLaune, 2008).

A description of decomposition based on first-order exponential decay is inadequate to capture non-linear responses such as respiration pulses at rewetting due to combined microbial reactivation and changes in substrate availability (Manzoni et al., 2020) or priming effects associated with substrate quality. Phenolic compounds can downregulate enzymes responsible for decomposing other carbon compounds, resulting in negative priming effects that inhibit overall decomposition (Freeman et al., 2001). Conversely, the allocation of labile carbon through roots can stimulate decomposer activity in coniferous-dominated soils (Jílková et al., 2022; Leppälammi-Kujansuu et al., 2014; Li et al., 2020), though in nutrient-rich sites like the one simulated, this effect may be less important than in nutrient-limited sites. While these contrasting priming effects have been reported, the overall response of soil organic carbon decomposition to root exudates in coniferous forests remains unclear (Gundale et al., 2024). Interestingly, measurements of carbon accumulation in drained forested peatlands have often been conducted in nutrient-poor sites using chamber methods with root trenching, which do not account for the effect of labile carbon allocated by roots on heterotrophic respiration (Hermans et al., 2022).

Generally, a model that explicitly represents the interactions between organic carbon substrates and microbial communities is desirable for exploring priming effects and the consequences of increased precipitation variability. However, in the context of this study, we suspect that the additional uncertainties in the microbial process parameterisation would decrease the benefit of a microbial-explicit model. Besides increasing process representation, peat decomposition models based on first-order kinetics could benefit from representing SOM as measurable pools, especially if field-based decomposition data for chemically distinct, measurable SOM pools, coupled with field-based carbon balance data, become more readily available.

Similarly, the way plant carbon is simulated has certain limitations. For instance, carbon allocation within plant compartments follows a simplified scheme in which water and nutrient availability do not directly influence root allocation. Yet, plants can increase carbon allocation to roots to enhance resource acquisition (Prescott et al., 2020). The model also simplifies wood dynamics. It assumes a fixed proportion between sapwood and heartwood, considering sapwood the only wood fraction that respires, while woody litterfall is modelled as a fixed proportion of total wood mass. For this reason, as trees age and wood tissue comprise a larger fraction of total plant biomass, both wood respiration and woody litterfall increase. In reality, the proportion between sapwood and heartwood is dynamic, changing as trees grow. Wood growth originates in the sapwood, which gradually transforms into heartwood. This gradual change in the proportion between sapwood and hardwood affects both respiration rates and litterfall patterns, a process not captured by the model's fixed allocation scheme. The model could be improved with a more dynamic representation of wood dynamics. However, the allocation rates and respiration costs of sapwood are not well understood and are difficult to measure, posing challenges for accurately parameterising a model given their importance in tree carbon dynamics (Metzler et al., 2024).

Furthermore, the current model formulation suggests a strong response to rising atmospheric $CO_2$ concentration, leading to notably high photosynthetic rates during the second rotation. This effect is driven by enhanced carbon assimilation and water use efficiency, resulting in greater growth. These responses have been both theorised and observed in forests exposed to elevated $CO_2$ levels (Donohue et al., 2017; Sigurdsson et al., 2013). However, there is uncertainty about the magnitude of these effects due to long-term acclimation and interactions with other environmental factors, such as increasing ozone concentrations

or changes in vapour pressure deficit driven by higher temperatures, which may offset or alter the benefits of elevated $CO_2$ (Gustafson et al., 2018).

Despite uncertainties, it is still useful to analyse the system under conditions of very high carbon uptake by trees, especially because the model neglects understory vegetation and its contribution to carbon assimilation. It has been estimated that in nutrient-rich, drained forested peatlands, grasses and mosses can dominate photosynthetic activity during the first 10 years of

a forest rotation (He et al., 2016). This exclusion may result in the model underestimating GPP and litter inputs during the early years. Nonetheless, as evidenced by the measurements in this study, drained conditions during the initial years of forest rotation are characterised by much larger carbon losses due to elevated decomposition of soil organic matter, particularly when ditch network maintenance lowers the groundwater level (Korkiakoski et al., 2019; Palviainen et al., 2022) and stimulates aerobic decomposition (Evans et al., 2016; Nieminen et al., 2018)

Lastly, we assumed that a constant fraction of wood is allocated to HWP compartments. Our assumption is that 65% of harvested wood has been used in other studies (Kasimir et al., 2018). However, this fraction varies based on wood quality (Jonsson et al., 2018; Profft et al., 2009).

The current model formulation of carbon dynamics, based on common representations embedded in other models, generally provides a reasonable platform for analysing peatland systems despite certain limitations. While this contribution focuses on

a drained forested site, the model structure is flexible and applicable to other conditions, such as waterlogged soils (not drained) and natural vegetation, including grasses and mosses, provided appropriate parameterisation of the vegetation submodel is implemented.

### 4.3 System boundaries and metrics of carbon exchange

Assessing net carbon exchanges—and, by extension, the potential climate impacts—in drained forested peatlands is inherently

dependent on the delineation of system boundaries and the selected evaluation metrics. This study proposes that the most effective system boundaries for assessing long-term carbon exchanges are those where inflows take the form of gaseous carbon uptake from the atmosphere and outflows take the form of gaseous carbon releases to the atmosphere. The ecosystem+HWP boundary used in this study serves as an approximation of this premise.

Within these boundaries, different metrics can offer divergent perspectives on climatic effects. Our analysis reveals that the

system accumulated more carbon than it released towards the end of rotations, resulting in a positive NCB. However, NCB fails to account for the temporal dynamics of carbon accumulation within the system (Muñoz et al., 2024). A small, constant

carbon gain over time can yield the same NCB as carbon dynamics characterised by substantial initial losses followed by substantial later gains within the analysis period. However, these scenarios may not have equivalent effects on climate.

The influence of carbon dioxide on climate change manifested through alterations in the planetary energy balance, depends on both the atmospheric $CO_2$ concentration and the residence time of each $CO_2$ molecule in the atmosphere (Joos et al., 2013). Consequently, when a system exhibits substantial carbon losses throughout the analysis period, only compensated towards the end, the interval during which accumulated losses exceeded accumulated gains can be interpreted as a period of negative effect on the climate, despite an eventual positive effect (i.e., accumulated losses became less than accumulated gains). Furthermore, designating the end of the rotation as the final point of the analysis period introduces bias, as any accumulated carbon in biomass is relatively quickly lost upon harvesting.

This temporal information is captured by ICS, providing a more comprehensive assessment of the climatic impact of specific carbon dynamics within a system. This is especially important in drained peatlands where high carbon losses at the beginning of a rotation are compensated only towards its end, leading to negative ICS. A substantial proportion of Fennoscandian drained forested peatlands are approaching stand maturity, prompting imminent management decisions (Lehtonen et al., 2023). A very negative but improving ICS may be a representative pattern for these systems; therefore, avoiding clear-cutting is crucial to prevent declines in this metric. It would be important to assess the effects of different management strategies on ICS, especially those that do not rely on clear-cutting, such as continuous forest cover (Laudon and Maher Hasselquist, 2023).

While the ICS provides valuable information for evaluating the climatic impact of a specific trajectory of carbon exchange, as demonstrated in the present study, it does not account for the varying warming effects associated with different types of carbon compounds exchanged (e.g., methane emissions under waterlogged conditions). Therefore, to assess alternative land use scenarios for forested drained peatlands, such as rewetting, a metric that incorporates both temporal dynamics and the warming effects of all GHGs, such as cumulative radiative forcing (Murphy and Ravishankara, 2018), would be ideal. Given the relatively fast release of carbon from HWP after harvesting and the potentially high release of the potent greenhouse gas $CH_4$ during the initial years of successful rewetting (Escobar et al., 2022), the effect on climate of combining clear-cutting and rewetting could take a long time to be compensated (Ojanen and Minkkinen, 2020).

## 5 Conclusions

The ForSAFE-peat model was able to realistically reproduce soil abiotic (temperature and GWL) conditions and annual net ecosystem productivity at the drained and forested, nutrient-rich peatland at Skogaryd in Southern Sweden. The model predicted a substantial increase in biomass growth in the future following higher temperatures and atmospheric $CO_2$ concentration, supported by higher precipitation and nitrogen mineralisation, and shows that even such a large increase in photosynthesis may not compensate for the large carbon losses caused by enhanced decomposition from drained peat. The results underline the importance of choosing the appropriate system boundary for carbon budget estimates and argue for a more holistic budget accounting for the ecosystem and the fate of the harvested biomass. The study also shows how accounting

for the temporal dimension of the carbon budget of a managed forest site can give fundamentally different estimates of the potential effect on climate warming. The study contrasts the NCB, which only focuses on book-keeping balances over a given period, with the more integrative ICS, which accounts for the time $CO_2$ resides in the atmosphere and indicates that the former may give misleading estimates of climatic implications. Based on the testing at Skogaryd, we show that even if the nutrient-rich site may appear as a net sink at the end of a forest rotation, its legacy effect on climate can remain negative given that much of the captured carbon was released in the atmosphere longer than it was fixed at the site, thereby producing a warming effect. We finally argue for a pragmatic adoption of dynamic modelling in estimating the effects of forest management on climate warming despite their limitation as illustrated here and underline the importance of broader ecosystem boundaries in these estimates as well as more representative indicators accounting for the temporal aspect of forest management on carbon residence.

**Appendix A: Model sensitivity analysis**

Model sensitivity analysis was performed to test the effect of uncertainty on the initial nutrient status and future precipitation level (Figure A1). A total of 9 scenarios were created based on a combination of 3 initial CN ratio scenarios and 3 precipitation scenarios for the years 2020-2088. The sensitivity analysis reveals that both water and nutrient availability regulate carbon dynamics. The higher NCB at the end of the simulation is associated with the high-nutrient, high-precipitation scenario. In comparison, the lower NCB is associated with the low-nutrient low-precipitation scenario. The NCB difference between these two scenarios was 9180 $g_C$ m$^{-2}$. Given that the ICS is the time-integrated NCB, the sensitivity analysis also reveals negative ICS across all scenarios.

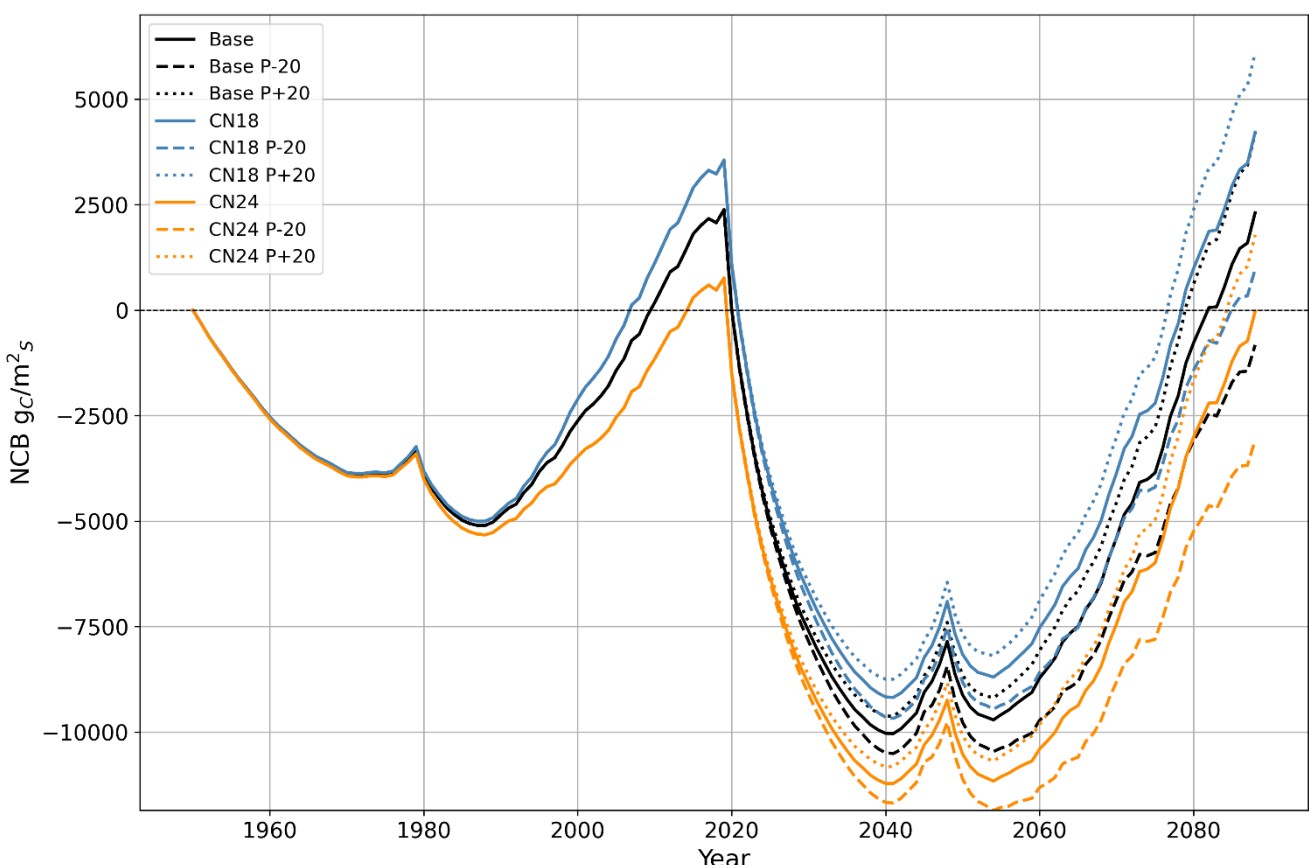

**Figure A1. Sensitivity analysis of net carbon balance (NCB) for the Ecosystem+HWP boundary: Each colour represents a different C:N ratio, and each line style indicates a specific future precipitation scenario. The black lines correspond to a C:N ratio of 21, used in the primary simulation of this study (Base). Blue lines represent a C:N ratio of 18, while orange lines represent a C:N ratio of 24. Dotted lines indicate a scenario with 20% higher precipitation from 2020–2088 compared to the main simulation, while dashed lines represent a scenario with 20% lower precipitation during the same period.**

Within the model formulation, nitrogen and water directly influence carbon dynamics, as illustrated in Figure A1. Nutrient content in soil organic matter regulates the nitrogen mineralisation rate, which controls nitrogen uptake by trees. This uptake determines leaf nitrogen content, thereby influencing GPP. Simultaneously, precipitation regulates soil water content, affecting both water uptake by trees and decomposition rates through soil water content.

Higher nutrient availability primarily impacts the net carbon balance (NCB) by increasing GPP. However, water availability may become a limiting factor as nutrient conditions improve and growth accelerates. In contrast, higher precipitation benefits the NCB by enhancing GPP and reducing decomposition rates. These causal relationships explain why similar NCB values were observed at the end of the simulation for two scenarios: one combining a C:N ratio of 21 with 20% higher precipitation

(dotted black line in Figure A1) and another with a C:N ratio of 18 (solid blue line in Figure A1) under precipitation levels matching the main simulation.

**Appendix B: Soil physical changes**

Peat soils are highly dynamic, undergoing expansion and contraction driven by changes in their carbon and water balance. To capture this behaviour, the model incorporates a dynamic volume approach, in which the soil organic matter balance directly controls each soil layer's thickness. This mechanism allows the model to simulate interactions between carbon accumulation, decomposition, and water content, which collectively influence the structure of peat soils over time. Figure B1 illustrates these

710 dynamics, highlighting the simulated thickness and bulk density changes throughout the analysis period.

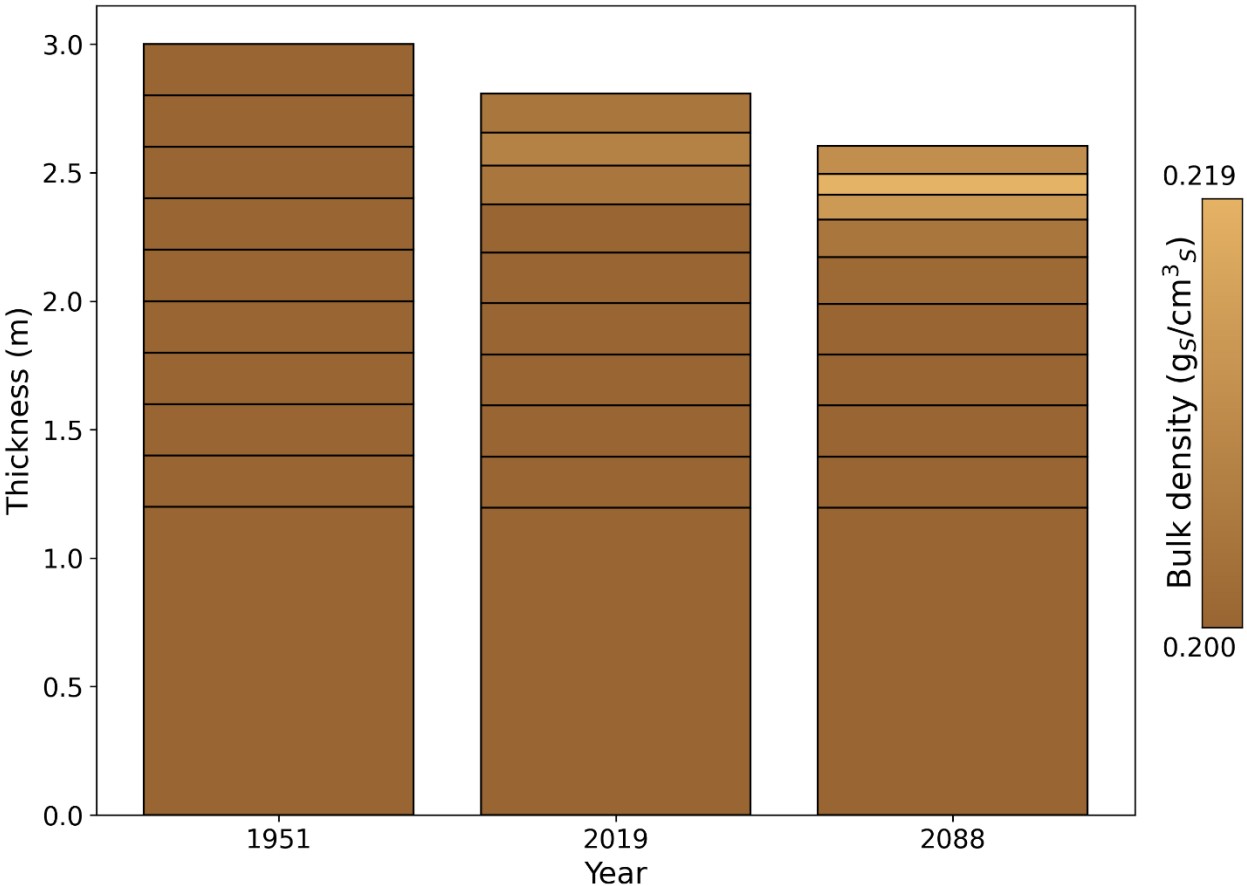

**Figure B1. Simulated peat soil thickness and bulk density at three key time points: 1951, 2019, and 2088. The year 1951 marks the beginning of the simulation, coinciding with the establishment of tree planting. By 2019, the first forest rotation is completed, followed by a second rotation ending in 2088. The y-axis represents the cumulative thickness of the peat soil layers, with the total thickness shown for each year. The colour gradient indicates the soil bulk density ($g_{soil}/cm^3_{soil}$) of the soil layers, where lighter shades represent higher bulk density values.**

Simulated changes in the soil profile followed observed patterns in drained peatlands. The overall thickness of the soil profile decreased due to a sustained negative carbon balance at the soil level, driven by higher peat decomposition rates compared to litter inputs. The reduction in thickness occurred in layers above the groundwater level, where aerobic decomposition dominates. Changes in bulk density were only noticeable in these upper layers. In the first layer, changes were less pronounced than in the second layer, as most litter inputs were concentrated in the first layer.

From 1951 to 2088, the model projected a total subsidence of 0.395 m, primarily driven by carbon losses in the upper four

layers of the peat profile. This result aligns closely with the estimated subsidence of 0.357 m in peatlands used for forestry over 136 years of drainage, as calculated using an empirically based model derived from a meta-analysis of centennial-scale shifts in the hydrophysical properties of peat induced by drainage (Liu et al., 2020).

Regarding changes in bulk density, the rate of change in the upper three layers was approximately $1.30\times10^4$ $g_{soil}$ $m^{-3}_{soil}$ $yr^{-1}$, which is lower than the $5.60\times10^4$ $g_{soil}$ $m^{-3}_{soil}$ $yr^{-1}$ estimated by Liu et al. (2020) for drained forested peatlands. Despite

differences between the original bulk density of those sites (0.07 $g_{soil}$ $cm^{-3}_{soil}$) and our initial bulk density (0.20 $g_{soil}$ $cm^{-3}_{soil}$), the underestimation of the rate of change is likely due to ForSAFE-Peat not accounting for the collapse of soil pore space under drained conditions. According to Liu et al. (2020), most changes in bulk density occur within the first 30 years of drainage, likely due to the collapse of macropores shortly after drainage (Silins and Rothwell, 1998).

In ForSAFE-Peat, bulk density changes are driven by the ratio of mineral soil content to organic soil content, which fluctuates

as organic soil content increases or decreases. If organic soil content decreases, the fraction associated with mineral soil content increases, meaning the average particle density also increases, as minerals are denser than organic matter. If the average particle density increases while porosity remains constant, bulk density increases.

**Appendix C: Further model performance evaluation.**

Further model evaluation was performed against temperature for depths of 0.15 m and 0.30 m. The modelled temperature at

740 0.20 m was similar to the observed temperature at 0.15m (Figure C1).

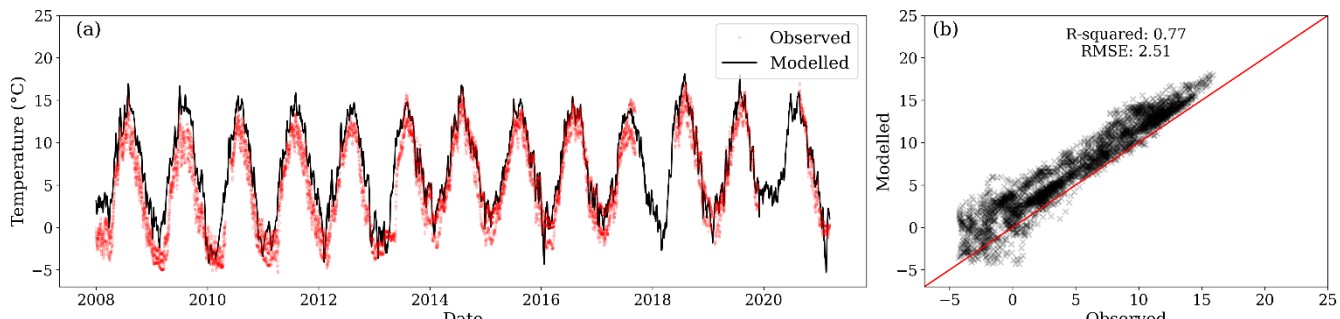

**Figure C1. (a) Modelled temperature for the second layer (black line) and observations at 0.15m depth (red dots) from three locations. (b) Relationship between observed and modelled values. During the period of comparison, the centroid of the first layer**

**was between 0.223m and 0.225m**

Equally, the modelled temperature for a depth of 0.38m was similar to the observed temperature at 0.30m (Figure C2). However, a slight overestimation is persistent at this depth.

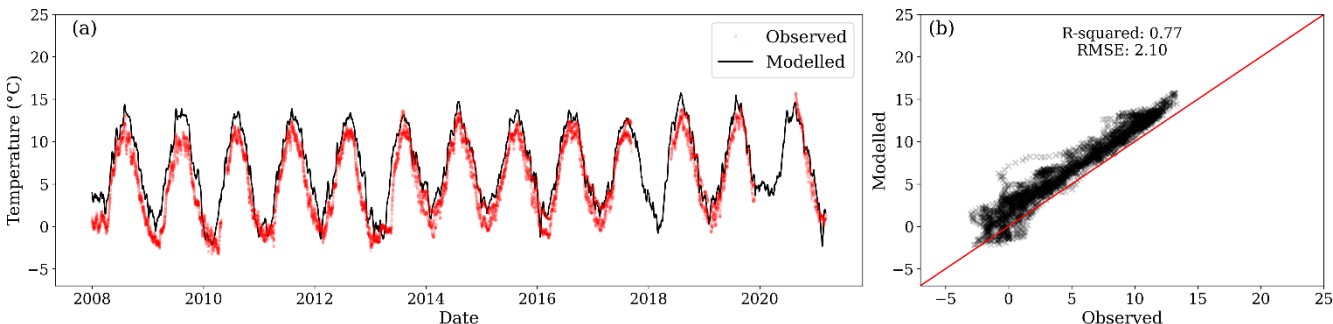

**Figure C2. (a) Modelled temperature for the third layer (black line) and observations at 0.30m depth (red dots) from three locations. (b) Relationship between observed and modelled values. During the period of comparison, the centroid of the first layer was between 0.372m and 0.364m**

## Code availability:

The original model code of ForSAFE-Peat is written in Fortran 90 and is freely available upon request to the model developers (see contact details above) with the intent to support new user in the initial stage of their work with the ForSAFE model.

## Data availability:

Field measurement data used to validate the model and yearly model outputs of carbon fluxes encompassing the full extent of the simulation are publicly available at 10.5281/zenodo.14831429

## Author contribution:

DE led the study. DE, SB and SM conceptualised the study. DE, SB and PV conducted the formal analysis and investigation. SM and JT assisted in the formal analysis and investigation. All the authors discussed the results together. DE wrote the original draft of the paper and produced the figures, with feedback from SB, SM and PV. All authors reviewed and commented on the original draft of the paper and its revisions.

## Competing interest:

The authors declare that they have no conflict of interest

## Financial support:

SM has received funding from the Swedish Research Council Vetenskapsrådet (grant 2020-03910) and the Swedish Research Council FORMAS (grant 2021-02121).

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
