# Peer review of "Evaluation of long-term carbon dynamics in a drained forested peatland using the ForSAFE-Peat Model."

_EGUsphere, 2024_

## Author Response (AR1)

Dear editor and referees,

We appreciate the opportunity to submit a revised version of our manuscript and are thankful for the feedback provided by the reviewers. In response to their comments, we have made several significant improvements to the manuscript. First, we added a sensitivity analysis as Appendix A, as proposed in our responses to Reviewers #1 and #2. This analysis explores the effects of different precipitation scenarios (which equate to varying water content) and different initial nutrient conditions, helping to test the model's behavior and accuracy. Additionally, we included another appendix that showcases the physical changes simulated by the model, such as bulk density and soil profile height, as suggested by Reviewer #1. This addition highlights a novel aspect of the model and strengthens its representativeness.

Furthermore, based on Reviewer #1's feedback, we updated the base run to the latest version of the model, which we have been refining for a future contribution analyzing various land use management options. This update resulted in a more realistic representation of woody litterfall rates, forest growth during the second rotation, and dissolved organic carbon (DOC) leaching, without significantly altering the values or affecting the study's conclusions. We also expanded on the rationale for the limited calibration, discussed the drivers for the second rotation in greater detail, and enhanced the discussion on model representativeness as a form of validation. Additionally, we included a description of the eddy covariance flux data used for the years 2020 and 2021.

In response to Reviewer #2, we have consistently used the term "forested drained peatlands" throughout the manuscript, reserving "afforestation" only when applicable. We also expanded the description of the ditch network to provide greater clarity. Addressing Reviewer #3's comments, we elaborated on the justification for using the ForSAFE model in the introduction and discussed its applicability in more detail. Finally, as suggested by Reviewer #4, we revised the supplementary materials to improve clarity regarding certain parameters and provided additional descriptions to indicate new developments made to the model.

These changes have significantly strengthened the manuscript, and we believe they address all the reviewers' concerns comprehensively. Below, we provide a point-by-point response to each reviewer's comments, with the original comments in red, our responses in black, and the revised text from the manuscript in green. For clarity, we have either directly displayed the revised text or indicated the specific lines or sections where the changes can be found, particularly when the revisions are too extensive to include in full. Reviewer comments are coded (e.g. R1C1 stands for reviewer # 1 comment # 1) and changes in the supplement are indicated by the code SPLMT.

**R1C1:**

*First, a full forest rotation approach has been undertaken earlier in He et al. 2016 Biogeosicences study for the same site, although use another model but the same conclusion was already made there. Moreover, Kasimir et al 2018 GCB further presented full rotational GHG balance (not only CO2 but also CH4, N2O) of their studied site for several land use scenarios, including final harvest of the forest. Thus, I would argue their findings on the system boundaries are already known. The advance of ForSAFE has a few interesting developments in it (e.g. simple dynamic volume for peat, although they did not discuss much about it), but put it in the community of peatland models, the novelty is rather minor.*

**Answer to R1C1:**

We appreciate the reviewer's comments regarding the novelty of our work and its relation to previous studies. While we acknowledge the significant contributions of He et al. (2016) and Kasimir et al. (2018), we believe our paper adds valuable insights to the ongoing discussion around northern peatland management, particularly in terms of system boundaries and model dynamics.

First, our study differs in the modelling approach and process representation. He et al. (2016) and Kasimir et al. (2018) noted that heterotrophic respiration decreased in the final 30 years of their simulated forest rotation, attributing this to a shallower water table and reduced peat "concentration" due to a static soil volume coupled with high decomposition flux. We address this issue by implementing a dynamic volume approach, which we believe better represents the coupling between soil organic matter decomposition and water table dynamics.

Second, the calibration strategies differ significantly. Both He et al. (2016) and Kasimir et al. (2018) utilized extensive site-specific calibration with approximately 30 parameters, which can achieve good fits to observational data but may do so for reasons that are not universally applicable. In contrast, our study emphasizes in mechanistic default parameterization, minimizing site-specific calibration to ensure broader applicability. For instance, we adopted the decomposition rate constant from long-standing models like ORCHIDEE (Qiu et al., 2019) and LPJ-GUESS (Chaudhary et al., 2017), as well as the spruce carbon assimilation from PnET model for (Aber et al., 1996). This approach allows us to reflect on assumptions used in widely accepted models and provides a foundation for testing the model under diverse conditions.

These methodological differences yield distinct results. For example, He et al. (2016) reported an average NEP of 217 gC/m$^2$ for 1980–2011, closely aligning with observed NEP during the dry year of 2008 (204 gC/m$^2$). However, using a mechanistic model and avoiding the need to calibrate our simulations, we observe that 2008 was an outlier regarding carbon uptake. This elasticity in response to environmental drivers is what allows us to simulate future NEP, but come with the tradeoff of a weaker fit to observations on high temporal resolution. Interestingly, in He et al. (2016), spruce NPP peaked in 1972 despite rising atmospheric $CO_2$ and temperature, while in our model, these better environmental conditions manage to modulate the effect of stand age. Further comparison is highly interesting, but model outputs of He et al. (2016) are not publicly available. We plan to make our model outputs available for future comparisons

Third, our definition and analysis of system boundaries differ markedly from those of Kasimir et al. (2018). While they present final values for various boundaries, the analysis in Kasimir et al. (2018) lacks a temporal perspective on how these metrics evolve. Moreover, the grouping of fluxes and, therefore their definitions of system boundaries differ from ours in several key aspects:

- Their system boundary, defined as soil, includes $N_2O$ and does not include carbon loss via water exports; in our work, we focus on carbon and include carbon loss via water.
- We do not consider any system boundary that is defined exclusively by fluxes included in the NEE.
- They do not incorporate the slow decay of a big proportion of harvested wood products (HWP), which we explicitly include in our "Ecosystem + HWP" boundary. This last point is particularly relevant for understanding post-clear-cut carbon dynamics and has not been incorporated in the previous analysis mentioned by the reviewer.

Additionally, our study simulates a second full forest rotation under climate change conditions—an aspect not explored in previous work. We agree that the historical management history of the site

was not entirely conventional, yet we chose to reproduce for a second generation in order to isolate the effect of a changing climate. This addition provides a more realistic "business-as-usual" scenario, knowing that the forest management plans are not always followed by the letter, and enabling us to test the model's performance under changing environmental conditions and assess the impacts on carbon dynamics. It also lays the groundwork for evaluating alternative land-use scenarios, such as peatland restoration through rewetting.

While we acknowledge that our model developments are incremental rather than revolutionary, the incorporation of a dynamic soil volume marks a significant improvement. In He et al. (2016), peat decomposition decreased over time despite rising temperatures, due to a static volume assumption. By allowing soil volume to change dynamically, we enhance the coupling between SOM decomposition and the water table, providing a more realistic representation of peatland processes.

We would like to take the opportunity to address the reviewer's concern about novelty to highlight that the dynamic volume approach is not included in previous studies. We would like to include the simulated change in physical properties (height of the soil profile and bulk density) in an appendix and discuss them as part of the representativeness of the model comparing with published data.

**Changes in text due to R1C1:**

Line 703 to line 736: **"Appendix B: Soil physical changes..."**

**R1C2:**

*Second, I have many concerns on their model evaluation, specifically: do three years of NEE data (one year under mature forest, two years after clear cutting) enough for constraining (or support) their full carbon budget analysis for the two-rotation periods? One-year of 2008 (note even a dry year thus not a normal climate year) NEE data to evaluate, or precisely speaking benchmark, 80 year (first rotation) of modelled C results, with a very detailed processed-based approach (multiple factors control the C flows) like ForSAFE. Not saying many of its simulated C flows did not benchmark with any field data at all. Their model data comparison (Fig. 5) shows large derivates between the measured and model daily data, suggesting these controls of the CO2 exchange are poorly captured by current model structure. The uncertainties in these (at least partly) also reflected by their simulated striking increase (double) growth over the second rotation period compared to the first rotation period (Fig. 6c). So, what cause these mismatches? if ForSAFE can not adequately simulate the underlaying process control and how these controls respond to clear cutting, as current results suggest. How can we trust the long-term predictions made by the model. This is a crucial question for the authors to think it over.*

**Answer R1C2:**

We acknowledge the reviewer's concerns regarding the evaluation of our model and the limitations of using short-term NEE data. Indeed, long-term carbon flux data from forested drained peatlands are scarce, with only one known site providing more than five years of eddy covariance (EC)-based NEE data. This scarcity of long-term measurements underscores the necessity of employing process-based models, as noted in our introduction, to explore carbon dynamics over extended periods.

We agree that the available NEE data is insufficient to constrain the full carbon budget across the two-rotation period. As explained in the previous point, our modelling approach does not require

constraining the model through calibration but rather to test the underlying mechanistic assumptions. We did not perform extensive calibration based on the NEE limited dataset. Instead, we dedicated a section of the discussion to comparing our model's outputs with published fluxes from relevant literature, demonstrating that the model still produces reasonable results. We argue that the strength of our paper lies in how we analyse and group these outputs to compare the relative importance of carbon gain and loss over time, providing new insights into carbon dynamics across different system boundaries.

Although limited data covers contrasting periods, including post-clearcut conditions, which offer valuable insights into soil responses with minimal vegetation influence, while the model does not perfectly capture daily fluxes, we must highlight that we deliberately avoided extensive calibration. The model involves numerous parameters to describe complex interactions within the system, and excessive calibration would risk overfitting, potentially obscuring structural deficiencies.

For the purpose of the paper, achieving a perfect fit to daily flux data was not a prioritised objective. Fine-tuning parameters such as decomposition rates, carbon use efficiency, or initial bulk density might improve fit but could mask the need for structural improvements, such as incorporating explicit microbial pools. Despite the mismatches at the daily scale, the model captures seasonal and yearly fluxes reasonably well. It also reproduces the magnitude difference between periods with and without tree cover, as shown in the results.

Given the long-term focus of our analysis, our priority was to produce reasonable yearly carbon flux estimates rather than optimise short-term flux predictions through extensive parameter adjustments. It is also important to note that uncertainties in the climatic time series, derived from reconstructions based on nearby weather stations, may contribute to discrepancies in daily flux values while still capturing overall trends.

Finally, the observed increase in growth during the second rotation is primarily driven by changing climate conditions. Rainfall increased by 23%, temperature by 40%, and $CO_2$ concentration by 57% during the second rotation. The enhanced growth reflects the influence of elevated $CO_2$ and temperature on photosynthetic rates, particularly under conditions with sufficient water and highlighting the fact that nutrient availability will remain high given the nutrient richness of the site. This effect is modelled using the carbon assimilation response function from the PnET model (Ollinger et al., 2002), which compiled empirical data on photosynthesis response to elevated $CO_2$.

To address the reviewer's comment, we clarified the rationale behind the lack of calibration and its implication for the model's performance compared to observations in the methods section. We expanded on the effect of the CO2 modifier function effect in the discussion, and we used the updated version the of the model as the base run.

**Changes in text due to R1C2:**

Line 86 – 87 in Introduction: By building on existing state-of-the-art models, ForSAFE-Peat is a suitable tool for exploring carbon dynamics in peatland systems and critically examining commonly used methods for their representation.

Line 88-89 in Introduction: This study used the ForSAFE-Peat model to conduct a long-term simulation spanning two complete forest rotations in a well-studied drained forested peatland in southwest Sweden, utilising primarily pre-calibrated parameters.

Line 219 to line 227 in Methods: ForSAFE-Peat calibration was intentionally limited, as the objective was to evaluate the outcomes of common modelling assumptions under the site's specific conditions that inspired our simulation. We manually calibrated two parameters: the modifier of the bottom layer hydraulic conductivity that controls percolation ($limK_{sat}$) and the fraction of wood that respires ($RWF$).

Line 401 to line 402 in results: During the second rotation, GPP notably increased to 1379 $g_C$ m$^{-2}$$_{soil}$ yr$^{-1}$, compared to 889 $g_C$ m$^{-2}$$_{soil}$ yr$^{-1}$ during the first rotation, so that the tree biomass at the end of the second rotation was 59% higher than in the first rotation..

Line 600 to line 608 in discussion: Furthermore, the current model formulation suggests a strong response to rising atmospheric $CO_2$ concentration, leading to notably high photosynthetic rates during the second rotation. This effect is driven by enhanced carbon assimilation and water use efficiency, resulting in greater growth. These responses have been both theorised and observed in forests exposed to elevated $CO_2$ levels (Donohue et al., 2017; Sigurdsson et al., 2013). However, there is uncertainty about the magnitude of these effects due to long-term acclimation and interactions with other environmental factors, such as increasing ozone concentrations or changes in vapour pressure deficit driven by higher temperatures, which may offset or alter the benefits of elevated $CO_2$ (Gustafson et al., 2018).

Line 41 SPLMT:…,MR (dimensionless) is the assimilation as a fraction of the morning rate.

**R1C3:**

*Third, the authors commented the first rotation period was non-conventional management, however, still design their second rotation periods with the same management. Why not have a simulation design with conventional management that additionally evaluate the sensitivity of the results to varying and more realistically forest management, because the chance of having a late 72% thinning and minor storm harvest two rotation in a row, is just way too low if not impossible. A model simulation with more realistic management would make more sense for forest management implications.*

**Answer R1C3:**

We acknowledge the reviewer's concern regarding using non-conventional management practices for both rotations in our simulations. Our rationale for maintaining the same management approach was to facilitate a direct comparison between the two rotations. This approach allows us to better evaluate the model's mechanistic representation of carbon and nutrient dynamics under similar conditions except for climate. Introducing a different management regime for the second rotation would have reduced the comparability between the two periods, making it more challenging to isolate and assess the effects of climate change and nutrient cycling dynamics.

That said, we recognise the importance of simulating more realistic management scenarios to enhance the practical applicability of our findings for forest management. Future studies could explore the model's sensitivity to varying management practices, providing insights into how these systems might respond under more conventional regimes.

**R1C4:**

*Please also note there are important issues concerning your data file shared on Zenodo: 1) the measured GWL data only contains the first two years; 2) the measured soil temperature data at 0.05 and 0.15m differ with what reported in your paper and clearly are wrong*

**Answer R1C4:**

The Zenodo file has been fixed, and the link has been updated in the manuscript.

**Changes in text due to R1C4:**

Line 755: Field measurement data used to validate the model, along with yearly model outputs of carbon fluxes encompassing the full extent of the simulation, are publicly available at 10.5281/zenodo.13629155

**R1C5:**

*Line 16 negative meaning uptake?*

**Answer R1C5:**

Metric fluxes and directions are explained thoroughly in the method section and figure 3, but further clarification has been added in the abstract.

**Changes in text due to R1C5:**

Line 16: Simulated carbon exchanges (a positive sign indicates gains and a negative sign indicates losses) were

**R1C6:**

*Line 37 gC*

**Answer R1C6:**

We are using another convention to be consistent with the way variables are described in the supplementary where the way of adding more information to units is with subscript; for example, cubic meter of water is $m3_w$ not m3W.

**R1C7:**

*Line 120 The site is agricultural used before tree planting how is the initial conditions for the soil consider those land use, does it make any difference or not. He et al 2016 clearly show how the initial conditions influence the results of the simulated GHG balance."*

**Answer R1C7:**

The initial conditions of the soil reflect its historical use as agricultural land, showing a relatively high bulk density compared to pristine peat and a very low CN ratio. These characteristics are indicative of the organic matter mineralisation that occurred during the site's agricultural phase. The initial

conditions used in the model were derived from on-site measurements of bulk density, ensuring that the land-use history of the site is appropriately accounted for in our simulations.

The sensitivity analysis previously mentioned explored initial CN ratio effects on the overall carbon dynamics of the site.

**R1C8:**

*Site description for the clear-cut site will also help the reader to understand the relevance of the understory vegetation in the C exchange for the forest few years of clear cutting*

**Answer R1C8:**

We appreciate the suggestion and included the following paragraph in the methods section to provide a detailed site description.

**Changes in text due to R1C8:**

Line 140-152: We simulated two forest rotations over the period from the beginning of 1951 to the end of 2088 at a drained afforested peatland located at Skogaryd Research Station (Klemedtsson et al., 2015) in the southwest of Sweden (58°23'N, 12°09'E). This site experiences a hemiboreal climate, has nitrogen-rich peat soil, features an effective drainage system, and is managed under conventional forestry practices. Originally an open fen valley, the site was drained in the late 19th century for agriculture before being converted to forestry in 1951. The ditch network forms a grid-like pattern, with the main ditch running north to south for 0.8 km, draining into Lake Skottenesjön. Smaller parallel ditches are spaced at varying distances. Until clear-cutting in 2019, the site was dominated by Norway spruce (*Picea abies*). The area affected by clear-cutting covered approximately 0.16 km², with logging debris left on most of the site (**Error! Reference source not found.**). Norway spruce was replanted on 2/3 of the site following clear-cutting. In 2022, a barrier was constructed in the main ditch to raise the water level in the northern third of the site. Visual inspections revealed that vegetation cover increased in the years following clear-cutting. By 2022, much of the site remained covered by logging residues, while grasses and sedges, particularly in areas without logging debris, reached heights of 90 cm in the middle of the summer.

.

**R1C9:**

*Line 115 section 2.2 site description. I generally lack the clear cutting description, is there any rewetting effects for the site, like blocking drainage ditches, do they replanting spruce trees or they left small trees to regrow? How is the hydrological state of the clear cutting site, also I am lacking the descriptions of the eddy tower data, since the measured EC data is not published before so a brief description of the set up and measure and process processing is of need.*

**Answer R1C9:**

See answer to R1C8. We also added a paragraph explaining the EC setup.

**Changes in text due to R1C9:**

Line 210 - 216: On-site NEP measurements were conducted in 2008 while trees were present, with subsequent data from 2020 and 2021 acquired post-clear-cutting, offering insights into soil respiration without substantial photosynthetic activity. NEP data processing and acquisition for the year 2008 is described in Meyer et al. (2013). For the years 2020-2021, the high-frequency data needed for flux calculations were acquired with an ultrasonic anemometer (USA-1, METEK GmbH, Germany) and a LI-7200RS gas analyser (LI-COR Biosciences, NE, USA) mounted at 2.15 m height above the low vegetation. The data acquisition frequency was 10 Hz, and the half-hourly average $CO_2$ flux was calculated with the EddyPro software, version 7.0.7 (LI-COR Biosciences, NE, USA) following the ICOS methodology (Sabbatini et al., 2018). Gaps in the dataset were subsequently filled using the REddyProCWeb online tool (Wutzler et al., 2018)..

**R1C10:**

*Line 173 why only these two parameters calibrated, even though we know the limKsat quite well. Plus, how these parameter uncertainties contribute to the uncertainties in the short-term model-data mismatch and long-term upscaling results?.*

**Answer R1C10:**

We calibrated only two parameters to avoid the clear risk of overfitting associated with adjusting a large number of model parameters. Our approach prioritizes ensuring that the main controls on carbon dynamics are not overly specific to this site's unique characteristics, thereby supporting the model's broader applicability.

Regarding the comment on how well limKsat is known, we are unclear about its intended context. Clarifications regarding the limited calibration can also be seen in the answer R1C2

**R1C11:**

*Figure 2, why not show the simulated fluxes with the width of the arrows*

**Answer R1C11:**

The figure was designed to illustrate the conceptual differences in system boundaries, as appears in the methods section before any results are introduced. Using arrow widths to represent simulated fluxes would shift the focus toward presenting results, which are discussed later in the manuscript. Keeping the figure schematic ensures clarity in explaining the methodology without prematurely referencing outcomes.

**R1C12:**

*Figure 3, there are clearly more than measured WTD time series, need to add that in the figure captions*

**Answer R1C12:**

If the reviewer refers to including several water table depth (WTD) measurement locations in the figure, we clarified this in the caption. The revised caption specifies that the figure includes data from multiple WTD locations to ensure accurate interpretation.

**Changes in text due to R1C12:**

Line 283-284: Figure 1. (a) Modelled GWL (black line) and observations (red dots) from 4 different locations within the site; negative values mean distance to the surface. (b) Relationship between the mean observed GWL (averaged across locations) and modelled GWL.

**R1C13:**

*I do have many concerns on how ForSAFE-peat handles hydrology which is arguably the most important variable for peatlands development and regulates C dynamics over this transitional clear-cutting period. The hydrological changes (including ET, Runoff etc) over the clear-cutting period and compare to before would be interesting to test the model in detail.*

**Answer R1C13:**

We agree with the reviewer on the critical role of hydrology in peatland carbon dynamics, particularly during transitional periods like clear-cutting. Our modelling objective is to simulate soil saturation and temperature conditions that are reasonable enough to test the coupled carbon and nitrogen dynamics in the system. Among hydrological variables, the position of the water table is arguably the most critical for peatland processes (Evans et al., 2021, Nature). Our results indicate that the model successfully simulates observed water table dynamics at the site, including the rise in water table levels following clear-cutting in the absence of ditch maintenance.

We acknowledge that excluding ground vegetation in the model may influence evapotranspiration values. However, the reasonable agreement in water table dynamics suggests that the model captures similar water balances, provided that precipitation data are accurate and lateral inflow is negligible. While some processes, such as swelling and shrinking, anisotropic conductivity, and pronounced hysteresis in water conductivity, are not included in ForSAFE-Peat or other models previously applied to this site, we believe that achieving reasonable water table simulations provides a valuable context for testing carbon dynamics. Furthermore, many components of the water balance (e.g. Run-off) were not measured, making the model test the reviewer suggests difficult.

**R1C14:**

*Figure 5 what is cold and warm period, define it and also why not show the daily measured vs daily modelled numbers*

**Answer R1C14:**

We focused on seasonal periods (warm and cold) rather than daily values to highlight that, despite daily discrepancies, the model successfully captures seasonality. This focus aligns with the time dimension and scope of the study, where capturing broader seasonal trends is more relevant to long-term carbon dynamics.

The definitions of warm and cold periods are provided in the text (lines 307–308) and have now been added to the figure caption for clarity.

**Changes in text due to R1C14:**

Lines 311-313: Figure 2. (a) Modelled net ecosystem productivity (black line) and observations (red dots). (b) Relationship between observed and modelled net ecosystem productivity values. Values for model evaluation correspond to the aggregation of fluxes into warm (May, June, July, August, September and October) and cold (November, December, January, February, March, and April) months of the year.

**R2C1:**

*Throughout the manuscript, the authors write about "afforested drained peatlands". Indeed, their site is afforested (with agricultural history) and their results primarily tell about an afforested peatland. But almost all the more general parts of the paper (and majority of the references also) actually address more generally forested drained peatlands. Some of those peatlands have been afforested, but many have been forested even in their natural state before drainage. I suggest to generally use "forested drained peatland", and only use "afforested drained peatland" only when truly addressing afforested peatland. "Afforestation" is well defined in forest-related fields of science (and also in common English), and it means turning a previously permanently open site into a forest*

**Answer R2C1:**

We appreciate the reviewer's comment regarding the terminology used in the manuscript. We agree that distinguishing between "afforested" and "forested" drained peatlands will enhance clarity. To address this, we will make the following changes:

- Throughout the manuscript, we now use the term "forested drained peatland" when discussing peatlands more generally and reserve the term "afforested drained peatland" specifically for instances directly related to our study site.
- We will also update the title of the manuscript to refer to the more general
- In the discussion, we will clarify that tree cover in northern peatlands increases with continentality. This will include noting that while peatlands with significant tree cover are rare in the UK, spruce and pine mires are more common in Sweden and even more prevalent in Finland.
- Additionally, we will add comments clarifying that our site is an afforested peat

These revisions aim to ensure that our terminology accurately reflects the context of our study and that our readers can differentiate between afforested and naturally forested drained peatlands.

**Changes in text due to R2C1:**

Lines 1-2: Evaluation of long-term carbon dynamics in a drained forested peatland using the ForSAFE-Peat Model.

Lines 12 -14: Management of drained forested peatlands has important implications for carbon budgets, but contrasting views exist on its effects on climate. This study utilised the dynamic ecosystem model ForSAFE-Peat to simulate biogeochemical dynamics over two complete forest

rotations (1951-2088) in a nutrient-rich drained peatland afforested with Norway spruce (*Picea abies*) in southwest Sweden.

Lines 141-142: We simulated two forest rotations over the period from the beginning of 1951 to the end of 2088 at a drained afforested peatland located at Skogaryd Research Station

Lines 249-251: By examining different system boundaries, we can offer diverse perspectives on the carbon exchanges (and thus the potential effect on climate) of drained forested peatlands

Lines 379-380: Focusing solely on the soil boundaries overlooks the primary mechanism through which drained forested peatlands accumulate carbon, which is the living tissue of trees..

Lines 496-504: It is important to note that our study site was actively afforested, as it was initially an open fen. While peatlands with substantial tree cover are relatively rare in the UK, spruce and pine mires are more common in Sweden and even more prevalent in Finland (Laine et al., 2006). Peatlands without prior tree cover before drainage are likely to have higher soil water saturation levels than those with substantial tree cover (Beaulne et al., 2021). On peatlands with substantial tree cover, carbon accumulation in the upper soil layers is likely more dependent on stabilisation mechanisms occurring under aerobic conditions (Kilpeläinen et al., 2023). In contrast, carbon in peatlands without substantial tree cover is more likely stabilised by anoxic conditions, making it more sensitive to water table drawdown. Such nuances in carbon dynamics are often lost in the broad categories used to account for carbon in these land-use systems..

**R2C2:**

*The model quite nicely follows the measured C balance variables. This is quite remarkable! This is very hard to achieve without some calibration of C process related parameters (because there is a super huge C storage in peat soil, which tends to decompose either way to fast or way too slowly in the model). Yet, based on the manuscript, only a couple of hydrology-related parameters where calibrated. Please, clearly tell if any of C process-related parameters were calibrated or not. And if not, please advertise and discuss how the measured C balance was so well followed by the model!*

**Answer R2C2:**

We also appreciate the reviewer's comment on the model calibration regarding the carbon balance variables. Indeed, we have tried to avoid extensive calibration of many parameters because our goal was to test common model assumptions related to Norway Spruce dynamics and peatland decomposition embedded within the ForSAFE-Peat structure. To address this comment, we expanded on the parametrisation of the model and clarified our calibration process:

- The calibration details are currently found in the methods. We added further explanation to clarify why we only calibrated the rate at which water leaves the bottom soil layer and the fraction of wood that respires.
- We calibrated against groundwater level observations and tree ring-derived biomass data available for the years 2008-2010. The rationale for this approach is to test the model outputs regarding carbon fluxes if forest stand dynamics and abiotic conditions (such as groundwater level (GWL) and soil temperature) are similar to those observed in the field. We added this

rationale and further explain how these calibrated parameters (fraction of wood that respires and hydraulic conductivity at the bottom soil) influence the results after lines 218-225.

- Additionally, we included in the results section highlighting that the model successfully captures the transition from carbon sink to carbon source, consistent with field observations.
- In the discussion, we mentioned that under a reasonable water regime, the decomposition rate constant used in ForSAFE—similar to that in other models like Orchidee and LPJ—yields realistic peat losses. Meanwhile, the PnET default parametrisation related to carbon assimilation, respiration and litterfall, together with the calibrated fraction of live wood, lead to reasonable tree biomass accumulation, resulting in an overall agreement with values reported in similar systems in the literature.

These additions will clarify our modeling approach and explain why the model well matched the measured carbon balance without extensive parameter calibration of soil organic matter decomposition and soil gas diffusion.

**Changes in text due to R2C2:**

Lines 219-227: ForSAFE-Peat calibration was intentionally limited, as the objective was to evaluate the outcomes of common modelling assumptions under the site's specific conditions that inspired our simulation. We manually calibrated two parameters: the modifier of the bottom layer hydraulic conductivity that controls percolation ($limK_{sat}$) and the fraction of wood that respires ($RWF$). $limK_{sat}$ directly controls water leaving the soil profile by modulating percolation, thereby affecting the soil water balance. $RWF$ influences autotrophic respiration, which affects the tree's carbon balance and, consequently, biomass. In turn, biomass impacts water uptake, influencing groundwater levels. The water table also affects biomass by controlling water availability and nitrogen mineralisation. Calibration of these two parameters was conducted by comparing model outputs to GWL observations from two locations (2008–2013) and to biomass estimates for the site (2008–2010) derived from tree ring data (He et al., 2016). Additional details are provided in Supplementary Information 2.

Line 308: The model successfully captured the site transition from a carbon sink to a source.

Lines 544-549: Under a reasonable abiotic regime, defined by realistic water table depth and soil temperature, the decomposition representation used in ForSAFE-Peat—similar to those of models like ORCHIDEE and LPJ—produces credible estimates of peat losses. Additionally, the PnET default parameterisation for carbon assimilation, respiration, and litterfall, combined with the calibrated respiring wood fraction, yields realistic tree biomass accumulation. These results align closely with values reported in the literature for similar systems, supporting the model's ability to simulate carbon dynamics in drained forested peatlands under comparable conditions.

**R2C3:**

*No sensitivity analysis is conducted nor sensitivity of results to model assumptions and parameter selection discussed. I am not saying that you should do some full-scale sensitivity analysis (might not be reasonable), but at least some discussion on how robust your results may be is needed! So please discuss sensitivity/robustness of the results*

**Answer R2C3:**

We agree with the reviewer that a sensitivity analysis could make results more robust. In response, we performed a sensitivity analysis. The sensitivity analysis was done on the initial conditions of nitrogen content, as our focus is on conditions with high nitrogen content, but field measurements indicate spatial variability in the carbon to nitrogen ratio (C:N), ranging between 18 and 24. This variability motivates us to explore how different initial nitrogen contents impact model outcomes.

To illustrate the response of the model to different water regimes, we run simulations under various future precipitation scenarios, adjusting precipitation levels by increasing and decreasing them by 20%. This will help demonstrate how water availability changes affect the peatland's long-term carbon dynamics. As in the first sensitivity analysis, also in this second one, we are motivated by the high uncertainty in projected precipitation and the potentially high impact of different precipitation regimes on the carbon cycle.

The sensitivity analysis was added as an Appendix.

**Changes in text due to R2C3:**

Line 679 to line 702: "**Appendix A: Model sensitivity analysis ...**"

**R2C4:**

*C loss from peat soil is a most central element of the drained peatland problem. You describe well the water table depth (the strongly controls C loss from peat), but the actual ditching (ditch depth, ditch spacing, both strongly impacting drainage both at the site and in the modelling world) is very poorly described. Also, how the ditch maintenance is carried out in the model? (indirectly you seem to say that ditch cleaning to 60 cm depth at stand regeneration, but please say clearly). Also how ditches develop, in addition to soil subsidence? They collapse, get filled with vegetation, etc. leading to faster decrease in ditch depth than what is caused by only soil subsidence). See e.g. these two studies on the subject: https://doi.org/10.46490/BF453; https://doi.org/10.14214/sf.10494, and better describe the ditch network, please!*

**Answer R2C4:**

We appreciate the reviewer's comment regarding the ditch network, as C loss from peat soil is a central aspect of drained peatlands. To address this, we made several modifications:

- We improved the site's description and added a figure where the ditch network is visible.
- We added a more thorough description of how the model considers drainage in the section "2.1 Model description"
- We explained better the specifics of the simulation for the ditching

**Changes in text due to R2C4:**

Lines 119-123: Additionally, specific layers can exchange water horizontally, simulating the impact of ditching on hydrological processes within the peatland. The ditch function is simulated by setting an initial drainage depth. Layers above this depth experience lateral outflow when water content exceeds field capacity, with outflow regulated by the layer's hydraulic conductivity and width, as described in Zanchi et al. (2021b). The drainage depth adjusts dynamically with changes in the soil

profile; when the soil profile height is reduced due to net losses of soil organic matter, the ditch depth is also reduced by the same magnitude.

Lines 144-148: Originally an open fen valley, the site was drained in the late 19th century for agriculture before being converted to forestry in 1951. The ditch network forms a grid-like pattern, with the main ditch running north to south for 0.8 km, draining into Lake Skottenesjön. Smaller parallel ditches are spaced at varying distances. Until clear-cutting in 2019, the site was dominated by Norway spruce (*Picea abies*). The area affected by clear-cutting covered approximately 0.16 km$^2$, with logging debris left on most of the site (**Error! Reference source not found.**).

Lines 185-194: We set the initial ditch depth at 0.6 m based on ditch depth estimations from previous work conducted at the site (He et al., 2016; Nyström, 2016). We aimed to simulate standard ditch network maintenance (DNM) practices. In reality, the ditch was not maintained after clear-cutting in 2019 due to a rewetting experiment that began in 2022. Therefore, NEP observations for 2020 and 2021 were made after clear-cutting and during a period without DNM. To integrate historical accuracy with our aim of representing conventional management practices, we reset the ditch depth to 0.6 in our simulation starting in 2022. In the model formulation, lateral drainage is influenced by changes in ditch depth, which reflect variations in soil profile depth and hydraulic conductivity due to changes in the bulk density of the layers susceptible to lateral drainage. In reality, ditch depth is also influenced by infilling caused by sedimentation, vegetation growth, and bank erosion (Hökkä et al., 2020). However, these processes are not incorporated into the model for the sake of simplicity. A more detailed description of the scenario parameterisation can be found in Supplementary Information 2.

**R2C5:**

*For the two C balance metrics, NCB has an unambigious interpretation and physical meaning (change in C storage). ICS on the other hand, although being mathematically solid (you can always integrate) and having a good goal (taking into account the residence time of C in the atmosphere) does not have an exact physical interpretation. This might not be a problem, if you didn't have better options. But you (excellent modellers) can very easily calculate radiative forcing, which in a physically sound way integrates C balance and residence time of C-gases in the atmosphere (your C-gases are virtually all CO2) into climate impact. And then you can look both at the time series of instantaneous radiative forcing and take mean of radiative forcing over study period. You have otherwise such an exemplary solid manuscript, that I think messing up with ICS that is "almost something" kind of spoils it.*

**Answer R2C5:**

We appreciate the reviewer's insightful comment regarding using the Integrated Carbon Stocks (ICS) metric. We recognize that the ICS may not be as intuitive as the Net Carbon Balance (NCB), and we acknowledge that explaining its importance clearly has been challenging.

We agree that calculating radiative forcing would provide a better understanding of climate impacts, and we are actively considering atmospheric-related metrics for a subsequent study where we will assess different land management alternatives. However, for this contribution, calculating radiative forcing would require an assessment of fluxes from an atmospheric perspective, which would involve altering the fluxes considered due to carbon leaching across the three system boundaries as well as harvested carbon in the ecosystem boundary. Additionally, it would necessitate changes in flux

conventions, which could create confusion—for instance, a carbon uptake viewed from the ecosystem's perspective is a positive flux, whereas from the atmosphere's perspective, it would be negative.

We believe that NCB (gC/m$^2$) is more closely related to radiative forcing (W/m$^2$), while ICS (gCyr/m$^2$) is akin to what some call cumulative radiative forcing (J/m$^2$) (Murphy & Ravishankara, 2018; https://www.pnas.org/doi/epdf/10.1073/pnas.1813951115). In essence, if we consider NCB from the perspective of atmospheric carbon, it represents the change in atmospheric carbon content over time, accounting for atmospheric carbon dioxide decay functions and other gas concentrations. This approach would allow us to calculate radiative forcing, effectively translating changes in carbon content into changes in radiative forcing (which has units of power). Integrating radiative forcing over time yields units of energy (power × time)

A particular value of NCB metric can be achieved through various pathways, and ICS provides a straightforward means to compare those pathways from a climate change mitigation perspective. The importance of the time aspect in assessing the climate benefits of carbon sequestration in ecosystems has been highlighted previously (Sedjo & Sohngen, 2012; https://doi.org/10.1146/annurev-resource-083110-115941). The use of a mass × time metric has also been proposed for accounting for carbon permanence in carbon accounting problems (Fearnside et al., 2000; https://doi.org/10.1023/A:1009625122628). As expressed in Muñoz et al. (2024), studies like Sierra et al. (2021; https://bg.copernicus.org/articles/18/1029/2021/) have shown that ICS can effectively account for the time carbon spends stored in ecosystems, providing a more comprehensive means of analyzing and comparing trajectories of carbon accumulation, as demonstrated in the figure from Muñoz et al. (2024).

[Figure]

**FIGURE 1**  Open in figure viewer  ⬇PowerPoint

Conceptual representation of the effects of implementing two different measures (A and B) to enhance SOC over a time horizon since the implementation of a measure to enhance SOC. (a) SOC stocks [*M*, mass units] and (b) soil carbon sequestration (CS) [*M T*, mass × time units] defined as the area under the curve of remaining carbon over time (integral of curves in panel a). Green arrows in (a) represent the total C sequestration as defined by Don et al. (2023) at times $t_1$ and $t_2$ for the measures A and B. CS$_{i,t}$ in (b) indicates the soil carbon sequestration of a measure *i* at a time *t*.

**Changes in text due to R2C5:**

Lines 236-246: The $ICS(T)$ is useful because it accounts for the time dynamics of carbon storage, which in turn control the cumulative contribution of a system to atmospheric cooling or warming (Muñoz et al., 2024; Sierra et al., 2021). When a system exhibits a very dynamic carbon exchange characterised by periods of large net losses and periods of large net gains, the $NCB(t)$ might vary between positive and negative. The interval of time during which accumulated losses exceed accumulated gains can be interpreted as a period of negative effects on climate, while the opposite is true for the interval of time during which accumulated gains exceed accumulated losses. The cumulative effect of fluctuations in carbon storage is captured by the $ICS(T)$ via integration of $NCB(t)$ throughout the time period from $t_0$ to $T$. ICS has been proposed to account for carbon permanence in a system (Fearnside et al., 2000). Studies like Sierra et al. (2021) have shown that ICS can effectively account for the time carbon spends stored in ecosystems, providing a more comprehensive means of analysing and comparing trajectories of carbon accumulation (Muñoz et al., 2024).

**R3C1:**

*However, there is no information or justification of applied model approach in the introduction. There are many models, which are used for this purpose, therefore authors should justify the selection of the approach applied in this study. It would be better to place the ForSAFE-peat model in the context of current modeling studies. The posed questions in the end of introduction can be answered depending on model features and complexity. If model is sufficiently complex and well parametrized, the answer on first question always will be yes.*

**Answer R3C1:**

We appreciate the reviewer's comment highlighting the need for a justification of the modeling approach in the introduction. We agree that the paper would benefit from placing ForSAFE-Peat in the context of current modeling studies and providing a rationale for its selection. To address this, we have added a new paragraph in the introduction which contextualizes ForSAFE-Peat among existing models and clarifies its features and relevance to this study. Additionally, we acknowledge the reviewer's concern about the first posed question, given that a sufficiently complex and well-parameterized model might inherently yield a positive answer. However, we emphasize that our study does not perform extensive calibration. Therefore, this question remains important for evaluating the ability of the model to capture system dynamics under the given parameterization.

**Changes in text due to R3C1:**

Lines 74-86: Here, we introduce the dynamic ecosystem model ForSAFE-Peat and use it to analyse long-term carbon dynamics in a drained forested peatland. ForSAFE-Peat builds on previous models of carbon dynamics in coniferous forest and peat soils. It simulates plant dynamics as a big leaf model where photosynthesis is a function of foliar nitrogen content as in the PnET model (Aber & Federer, 1992). This representation has been widely use to study managed coniferous forest in northern latitudes (Belyazid et al., 2011; Belyazid & Zanchi, 2019; de Bruijn et al., 2014; Gustafson et al., 2020). ForSAFE-Peat simulates the soil as a set of layers that can expand or contract due to soil organic matter content changes, similar to peat development models like HPM (Frolking et al., 2010). Soil organic matter is represented by several compartments, including litter that, during decomposition, provides carbon and nutrient inputs to peat pools, resembling approaches like the one implemented

in Yasso07 (Didion et al., 2014). This allows a simple representation of litter quality and peat. Decomposition is described as a first-order exponential decay process where the peat decomposition rate constant is the same used to evaluate future carbon dynamics of northern peatlands by land-surface models such as ORCHIDEE (Qiu et al., 2018) and LPJ-GUESS (Chaudhary et al., 2022). By building on existing state-of-the-art models, ForSAFE-Peat is a suitable tool for exploring carbon dynamics in peatland systems and critically examining commonly used methods for their representation.

Lines 87-95: In this study, we used the ForSAFE-Peat model to conduct a long-term simulation spanning two complete forest rotations in a well-studied drained forested peatland in southwest Sweden, utilising primarily pre-calibrated parameters. Model outputs were analysed to represent various system boundaries and different metrics were applied to evaluate carbon exchanges across these boundaries. While acknowledging the potential significance of $N_2O$ emissions in drained fertile peatlands (Jauhiainen et al., 2023), we focused on carbon dynamics. Consequently, we explore the following two questions:

i.    How well does ForSAFE-Peat reproduce field-based observations related to carbon dynamics in a northern drained forested peatland?

ii.   How do patterns of modelled carbon exchange vary across different system boundaries in a northern drained forested peatland?

**R3C3:**

*Concerning the applied forest model: it was tested only for forest site. In fact, more general approach for simulating C dynamics in drained (and not drained and restored) peatland would be preferable, where other land use options also considered like grassland and even arable land. The applicability of used model in broader sense should be at least discussed.*

**Answer R3C3:**

We appreciate the reviewer's insightful comment regarding the broader applicability of the model to simulate carbon dynamics under various land use scenarios, including grassland and arable land, and the potential for its use in restored peatland systems. We agree that this reflection was missing and have addressed it in the revised manuscript. We acknowledge the need to discuss the general applicability of the ForSAFE-Peat model to broader ecological contexts. We are actively adapting the model to simulate other plant functional types, allowing it to represent different ecological states. Importantly, the soil dynamics associated with a restored peatland are expected to emerge naturally from the current model formulation. To reflect on this, we added these ideas to the discussion.

**Changes in text due to R3C3:**

Lines 617-621: The current model formulation of carbon dynamics, based on common representations embedded in other models, generally provides a reasonable platform for analysing peatland systems despite certain limitations. While this contribution focuses on a drained forested site, the model structure is flexible and applicable to other conditions, such as waterlogged soils (not drained) and natural vegetation, including grasses and mosses, provided appropriate parameterisation of the vegetation submodel is implemented.

**R3C4:**

*Reference list for the literature sources mentioned in the supplement with model description need to be provided*

**Answer R3C4:**

Indeed we agree, it has been added.

**Changes in text due to R3C4:**

Line 500-645 SPMLT: References…

**R3C5:**

*Figure 3a – it is not clear, which data are shown – observations from different locations, averaged observations or observations from one of the locations. Please extend the figure caption accordingly.*

**Answer R3C5:**

Figure 3 and figure 4 captions have been updated to clarify that observation are from different locations within the site of interest

**R3C6:**

*Discussion section 4.1. The authors mention in L407 that soil emission factors can fall into several climatic and nutrient categories, but do not report which category the site in question falls into. It might be useful to mention that all published data taken for comparison with model predictions belong to the specific category. In this relatively long section, the authors look at various model results and compare them with various published data sets. The main message is somewhat blurred in this type of presentation. Perhaps a summary of all cited data in the table with a parallel presentation of the model prediction made in the current paper would be useful.*

**Answer R3C6:**

We appreciate the reviewer's comment and agree that the discussion could be clarified further. To address this, we have added a conclusive sentence at the end of the relevant paragraph

Regarding the suggestion to include a table summarizing all cited data alongside the model predictions, we opted not to include a table for the following reasons:

- The study reports different values for the same variable (e.g., GPP) at different points during the forest rotation, reflecting temporal variability that aligns better with the field-based data we are comparing against.
- For example, some GPP values are presented for specific periods when LAI (Leaf Area Index) was comparable to field observations, while others are presented for two years post-clear-cutting to match the context of specific reported field measurements.

- Creating a table would require including substantial supplementary information to explain the context and temporal variability behind each data point, which we believe is more effectively communicated through text.

**Changes in text due to R3C6:**

Line 478-489: Site conditions can substantially influence the magnitude of carbon fluxes in northern drained forested peatlands. Therefore, soil emission factors for this land category are classified based on nutrient availability, climate conditions, and drainage level (Jauhiainen et al., 2023; Wilson et al., 2016).

In our simulated peatland, nutrient conditions are primarily determined by an initial soil organic matter C:N ratio of 21. Under these conditions, sites are often classified as *Herb-rich* type (Ojanen et al., 2010) or eutropic (Minkkinen et al., 2020). The average modelled soil temperature of 7.0°C from 1990 to 2020 aligns with values observed in cool temperate or hemiboreal sites in southern Sweden and Estonia (Minkkinen et al., 2007; Ranniku et al., 2024).

Regarding drainage, during the first rotation, the mean annual GWL was -0.43 m—comparable to other well-drained forested peatlands (Leppä et al., 2020; Maljanen et al., 2012; Menberu et al., 2016). Slightly lower values were observed toward the end of the rotation due to peat subsidence. Reported mean annual GWL values typically range between -0.3 m and -0.5 m at distances of 5 m and 15 m from the ditch, respectively (Haapalehto et al., 2014). Water table fluctuations simulated by ForSAFE-Peat reflect those of a well-drained site with functional ditches.

**R3C7:**

*Discussion section 4.2, L481-490. The authors admit that linear decomposition rate constants are not adequate to reflect the real processes, but they also defend the approach used, saying that the model representativeness of the measurements is satisfactory. Where is the truth, and what would the authors ultimately recommend? I assume that the use of multiple pools, as used in the current SOM decomposition concept, gives the model the necessary flexibility even for linear decomposition, but, unfortunately, there are no data to prove this, e.g. for cellulose, lignin and peat decomposition*

**Answer R3C7:**

We appreciate the reviewer's insightful comment regarding the limitations of linear decomposition rate constants and the representativeness of the current approach. To address this, we have clarified and expanded the discussion to highlight the model's applicability and limitations and provide recommendations for future improvements.

**Changes in text due to R3C7:**

Lines 562-567: The model followed commonly used formulations for peat soils (Kleinen et al., 2012; Qiu et al., 2018), where peat is defined as a conceptual compartment with unspecified chemistry that decomposes following first-order exponential decay, with rate constants modified by environmental conditions. Even though this description provides reasonable carbon dynamics at yearly and decadal time scales, limitations within this representation might explain why the model did not represent daily NEP fluxes well. Firstly, in the current model structure, decomposition rates increase slowly

with moisture content until field capacity, but this response could be faster in peat soils (Rewcastle et al., 2020; Ťupek et al., 2023).

Lines 580-585: Generally, a model that explicitly represents the interactions between organic carbon substrates and microbial communities is desirable for exploring priming effects and the consequences of increased precipitation variability. However, in the context of this study, we suspect that the additional uncertainties in the microbial process parameterisation would decrease the benefit of a microbial-explicit model. Besides increasing process representation, peat decomposition models based on first-order kinetics could benefit from representing SOM as measurable pools, especially if field-based decomposition data for chemically distinct, measurable SOM pools, coupled with field-based carbon balance data, become more readily available.

**R3C8:**

*L539: It is not entirely clear, what alternatives other than continuous forest cover were considered. Can your modelling approach be used to infer the applicability of alternative land uses, how should it be expanded? And the following statement... How can the full GHG budget (CH4 and N2O) can be considered? Is its consideration critical for the current study? If so, then the conclusions are somewhat vague...*

**Answer R3C8:**

We appreciate the reviewer's comment and take this opportunity to clarify our approach and conclusions. Our study demonstrates that clear-cutting significantly affects the carbon metrics used, making avoiding clear-cutting a reasonable conclusion if the objective is to improve these metrics. While other management options exist, we highlight continuous forest cover (CCF) as a strategy worth exploring given that CCF avoids clear-cutting. To make this reasoning more explicit we revised the paragraph in line 642.

We intend to discuss the full GHG budget's limitations and applicability. For the current study, the ICS is useful as it provides a metric to assess the impact of the simulated site, with negative values being undesirable from a climate change mitigation perspective. This makes it suitable for evaluating a single trajectory of carbon exchange and comparing system boundaries. However, its applicability is limited for comparative analyses between land-use scenarios. To clarify this point, we revised the paragraph in line 647.

**Changes in text due to R3C8:**

Line 645-651: This temporal information is captured by ICS, providing a more comprehensive assessment of the climatic impact of specific carbon dynamics within a system. This is especially important in drained peatlands where high carbon losses at the beginning of a rotation are compensated only towards its end, leading to negative ICS. A substantial proportion of Fennoscandian drained forested peatlands are approaching stand maturity, prompting imminent management decisions (Lehtonen et al., 2023). A very negative but improving ICS may be a representative pattern for these systems; therefore, avoiding clear-cutting is crucial to prevent declines in this metric. It would be important to assess the effects of different management strategies on ICS, especially those that do not rely on clear-cutting, such as continuous forest cover (Laudon & Maher Hasselquist, 2023).

Line 647-653: While the ICS provides valuable information for evaluating the climatic impact of a specific trajectory of carbon exchange, as demonstrated in the present study, it does not account for the varying warming effects associated with different types of carbon compounds exchanged (e.g., methane emissions under waterlogged conditions). Therefore, to assess alternative land use scenarios for forested drained peatlands, such as rewetting, a metric that incorporates both temporal dynamics and the warming effects of all GHGs, such as cumulative radiative forcing (Murphy & Ravishankara, 2018), would be ideal. Given the relatively fast release of carbon from HWP after harvesting and potentially high release of potent $CH_4$ during initial years of successful rewetting (Escobar et al., 2022), the effect on climate of combining clear-cutting and rewetting could take long to be compensated (Ojanen & Minkkinen, 2020).

**R3C9:**

*L 24: …consistent with the…*

**Answer R3C9:**

Fixed now in line 26

**R3C10:**

*L 30-31: A bit strange selection of three countries. What is about others? How big this coverage in respect to all drained peatlands in Northern Europe?"*

**Answer R3C10:**

We revised the sentence to have a better flow from general to the particularity of the study

**Changes in text due to R3C10:**

Line 31-33: Forestry on drained peatlands is a widespread land management practice in the northern hemisphere, covering approximately 15 million hectares, and it has important implications for carbon budgets (Leifeld et al., 2019). This practice is widespread in Fennoscandia, spanning around 5.7 million hectares in Finland and 1.5 million hectares in Sweden (Vasander et al., 2003)

**R3C11:**

*L 58: relevance (?)*

**Answer R3C11:**

Fixed from relevant to relevance now in line 61

**R3C12:**

*L65: Mamkin et al is missing in reference list*

**Answer R3C12:**

Fixed

**Changes in text due to R3C12:**

Line 930: Mamkin, V., Avilov, V., Ivanov, D., Varlagin,...

**R3C13:**

*L75: I would suggest to formulate the first research question a bit differently, say more specific. For example, Do calibrated ForSAFE-Peat model capable to describe field-based observations....*

**Answer R3C13:**

See next answer.

**R3C14:**

*L77: Second research question also can be more specific. It is not clear what exactly you mean with system boundaries."*

**Answer R3C14:**

To introduce what we mean by system boundary we modified a paragraph in the introduction as follows:

**Changes in text due to R3C14:**

Line 87-95: In this study, we used the ForSAFE-Peat model to conduct a long-term simulation spanning two complete forest rotations in a well-studied drained forested peatland in southwest Sweden, utilising primarily pre-calibrated parameters. Model outputs were analysed to represent various system boundaries and different metrics were applied to evaluate carbon exchanges across these boundaries. While acknowledging the potential significance of $N_2O$ emissions in drained fertile peatlands (Jauhiainen et al., 2023), we focused on carbon dynamics. Consequently, we explore the following two questions:

    i.    How well does ForSAFE-Peat reproduce field-based observations related to carbon dynamics in a northern drained forested peatland?

    ii.    How do patterns of modelled carbon exchange vary across different system boundaries in a northern drained forested peatland?

**R3C15:**

*L124: Maybe add information about RCP 6.0 – e.g. moderate temperature increase*

**Answer R3C15:**

We added information about the RCP 6.0

**Changes in text due to R3C15:**

Line 162-164: RCP 6.0 represents a medium stabilisation pathway, where greenhouse gas emissions peak around 2080 and decline thereafter, reflecting a future with moderate climate change mitigation efforts.

**R3C16:**

*References: Sierra (2024). Please do not cite unpublished work. Check if it is accepted and delete if not at the moment of your publication.*

**Answer R3C16:**

The reference has been eliminated and changed for Muñoz et al 2024

**R3C17:**

*EDC – is not a common abbreviation, try to avoid it and at least explain at the moment of first appearance."*

**Answer R3C17:**

The acronym is now defined before first appearance

**Changes in text due to R3C17:**

Line 364: easily decomposed compounds (EDC), cellulose…

**R3C18:**

*L302: It is not fully clear what you mean with the decreasing availability. Peat has always the same properties? Maybe explain this shortly here and not in the supplementary model description.*

**Answer R3C18:**

We appreciate the reviewer's comment and take this opportunity to clarify. Linear kinetics make the decomposition flux dependent on the amount of substrate available. In the second rotation, the first two soil layers experienced substantial decomposition, reducing the peat mass and, consequently, the amount of peat available for further decomposition. However, this reduction was compensated by higher decomposition rates, primarily driven by increasing soil temperature. To address this point in the manuscript, we revised the sentence.

**Changes in text due to R3C18:**

Line 364-366: The decreasing availability of peat in the first three soil layers, resulting from reduced peat mass due to decomposition, did not lead to lower decomposition fluxes because increasing soil temperature promoted decomposition

**R3C21:**

*L349: losses*

**Answer R3C21:**

Fixed throughout the text.

**R3C22:**

*L483: It is well established…*

**Answer R3C22:**

Fixed

**R3C23:**

*Table 1: Full names for the NCB – net carbon balance and ICS- integral carbon storage should be given in the title. I would use the same dimension for ICS in all three system boundaries, namely 10^6. It helps to see immediately the difference.*

**Answer R3C23:**

We agree, Table 1 has been fixed accordingly.

**R3C24:**

*Figure 6: delete 'plots' in caption. I suggest to plot stacked areas for the combined figs a and b and for combined figs c and d. HWP in the plot description can be fully spelled.*

**Answer R3C24:**

We agree, figure 6 has been fixed according to the comment

**R3C25:**

*L285: Not clear what is said: CO2 emission is the most significant and then is stated that leached DOC contributed 10% and CO2 -4%, i.e. less than DOC.*

**Answer R3C25:**

We meant leached CO2 dissolved in water, now is corrected in the sentence

**Changes in text due to R3C25:**

Line 347-348: The most substantial outflow was through soil $CO_2$ emissions, whereas leached carbon (DOC, $CH_4$ in water and $CO_2$ in water) contributed only 15% of total outflows.

**R3C26:**

*L511: what is DNM?*

**Answer R3C25:**

Fixed, now we introduce the acronym first.

**Changes in text due to R3C26:**

Lines 185: We aimed to simulate standard ditch network maintenance (DNM) practices.

**R4C1:**

*The authors use a pre-existing and tested model for simulating tree stand development and carbon dynamics, and incorporate a new description for organic soils, describing their carbon dynamics. The model is extensively described in the supplement. However, it was not immediately clear to me which parts of the described model are already published, and what is new here. This makes it hard to assess the work presented. I think the supplement would need a more clear distinction between these components*

**Answer R4C1:**

The rationale behind the supplement was twofold: first, to provide a detailed explanation of how carbon dynamics are represented in the model, as this is the primary focus of the paper; and second, to highlight aspects of the model that have been improved for this contribution. However, we recognise that the distinction between previously published components and new developments was not made sufficiently clear. To address this, we improved the first paragraph of the detailed model description in the supplement, explicitly clarifying which features are pre-existing and which are novel to this contribution. Additionally, incorporated clear statements during the mathematical descriptions to indicate where new features are introduced

**Changes in text due to R4C1:**

Lines 3-21 SPLMT: A general description of the previous version of the ForSAFE model can be found in Wallman et al. (2005). The model simulates plant dynamics based on the PnET-CN model (Aber et al., 1997), soil chemistry based on the SAFE model (Alveteg et al., 1998; Warfvinge et al., 1993), water dynamics based on the HBV/PULSE model (Andersson, 1988; Bergström, 1991) and soil decomposition dynamics based on the DECOMP model (Wallman et al., 2006; Walse et al., 1998). The model was further developed to include daily dynamics among other processes, such as lateral water movement, by Yu et al. (2018) and Zanchi et al. (2021).

In this contribution, we refined the plant's carbon allocation and respiratory processes. Given different turnover and growing rates between foliage and roots, allocation to roots is constantly recalculated to follow foliage demand. Maintenance respiration is decoupled from photosynthesis and is now a function of plant tissue biomass and temperature. To better capture soil dynamics in

peats, this version introduces a dynamic soil volume approach and considers the anaerobic decomposition of soil organic matter. Furthermore, soil nitrogen mineralisation and mineral nitrogen transformations have been changed to recreate better soil microbial processes. Soil organic matter decomposition is coupled with a fixed microbial carbon to nitrogen ratio to calculate mineralisation and immobilisation. In contrast, mineral nitrogen is transformed through nitrification and denitrification with explicitly modelled microbial groups. The description of soil temperature dynamics has been improved by solving the heat equation tailored for peat soils.

While this description primarily focuses on carbon dynamics, other subcomponents are briefly explained, emphasising changes made in this contribution. Carbon is represented as a set of compartments that denote different states of carbon within the system, such as carbon within foliage biomass, labile carbon allocated to roots, cellulose-like carbon in the soil, and so forth. Figure S1 summarises a schematic of organic carbon compartments within the model.

Examples of clarifications of new developments:

Line 127 SPLMT: For this contribution, we included the specific tissue's growth respiration cost,...

Line 147 SPLMT: The calculation of maintenance respiration was changed for this contribution

Line 184 SPLMT: For this contribution, anaerobic decomposition dynamics were included in the model formulation.

**R4C2:**

*Also, more references would be needed: now many parameters were given, but with no source. Or at least clarify the source. For example, does Equation (59) in the supplement come from Manzoni & Porporato (2009), referenced in the previous sentence? Same applies to (at least, non-exhaustive list) lines 402 -- 404, 422 -- 423, 440 and the last three lines of Table S8*

**Answer R4C2:**

To address this concern, we propose to be more explicit in indicating which equations are new to the model and their sources. This aligns with our response to R4C1, as both relate to improving clarity in the supplement. Regarding parameter sources, we recognize that linking parameters more directly to the equations in which they are used will enhance clarity

**Changes in text due to R4C2:**

Lines 313-316 SPLMT: Nitrogen mineralization (denoted by *NM*) from soil organic matter decomposition produces ammonium and is associated to the C:N ratio of the decomposition flux and the microbial biomass C:N ratio (Manzoni & Porporato, 2009). We have adapted the previously mentioned principle to both aerobic and anaerobic decomposition flux to calculate net nitrogen mineralization with the following equation:

Line 412-420 SPLMT: Variables used for plant carbon dynamics mentioned in the model description are grouped in Table S1, while parameters and their sources are grouped in Table S2. Most parameters come from articles presenting or applying the PnET model; however, we include new parameters associated with the new developments previously mentioned. Maintenance respiration rate constants ($MR_{TF}$, $MR_{TB}$, $MR_{TW}$, and $MR_{TR}$) were obtained from Metzler et al. (2024), these

parameters are necessary in equation 29. Also, a maximum root growth rate constant is introduced for Equation 18, with a value of 0.05 d-1 derived from the upper range reported for Norway Spruce in a nutrient manipulation study (Sell et al., 2022). As mentioned in section 1.1, the parameter respiring wood fraction (RWF) modulates the maintenance respiration of the woody tissue. The value was obtained by manually calibrating against proxy variables observations related to biomass size (tree ring data from 2007-2009 and GWL data from 2008-2013).

Line 440-456 SPLMT: Variables used for soil carbon dynamics are grouped in Table S5, while parameters and their sources are grouped in Table S6. Most parameters were derived from the original DECOMP model (Wallman et al., 2005; Walse et al., 1998) and recent modifications to ForSAFE (Yu et al., 2018). For this contribution, the previously named recalcitrant SOM compartment was renamed the peat compartment. The aerobic decomposition rate constant was taken from Clymo et al. (1998). This version of the model incorporates anaerobic decomposition based on first-order kinetics, as shown in equation 33. Therefore, we added to anaerobic decomposition rate constants ($PKan_i$) for different soil organic matter compartments. For example, the anaerobic decomposition rate of the EDC compartment (PKan$_{SE}$) has been assumed to be 10% of its aerobic rate, while the anaerobic rate of the cellulose compartment (PKan$_{SC}$) is assumed to be 1%. based on ranges for anaerobic decomposition of polysaccharides in incubation studies (Benner et al., 1984). Limited lignin decomposition is assumed to occur anaerobically; the rate constant (PKan$_{SL}$) is assumed to be 0.01%, informed by Reuter et al. (2024). For peat, the anaerobic decomposition rate constant (PKan$_{SL}$) is derived from catotelm decomposition rate constants (Clymo et al., 1998). Although anaerobic decomposition rate constants are less relevant under drained conditions, they are critical when the model is applied to waterlogged conditions.

The model representation of decomposition does not include an explicit representation of soil microbes in organic matter decomposition. However, microbial contribution to carbon stabilization is implicitly captured via parameters representing the carbon fraction that is not mineralized ($1 - M_{an-CO_2} - M_{an-CH_4}$). The value for the fraction that is not mineralized and goes back into peat from the peat compartment itself ($\gamma_{j\text{-SP}}$) is derived from the value of carbon assimilated into microbial biomass in Yu et al., 2018.

Line 464-466 SPLMT: Variables used for soil gaseous carbon dynamics are grouped in **Error! Reference source not found.**, while parameters and their sources are grouped in **Error! Reference source not found.**. Parameters associated with plant-mediated gas transport are not applicable to Norway spruce due to the absence of aerenchyma tissue in its root system. This explains the assigned values for parameters AP, TRτ and RCS in table S8.

**R4C3:**

*Related to that, the performance of the peat module may need more validation. Also, I found some pecularities about the peat fluxes: for example, seems that peat is formed from the other SOC pools. Is this realistic?*

**Answer R4C3:**

To further explore and validate the model, we conducted a sensitivity analysis by altering nutrient content and water availability. The results of this analysis are included in Appendix A in the main manuscript, as mentioned in answers to previous comments. Additionally, we provided an appendix showcasing the physical changes in the peat profile simulated by the model (e.g., bulk density and thickness changes). These simulated results are compared with published data to offer an additional layer of validation for the peat module, which we believe will strengthen confidence in the model's performance.

Regarding the question about peat fluxes, we argue that the current model description reflects a reasonable assumption grounded in field observations. Peat formation can be conceptualised as a humification process where organic material is transformed through microbial activity and physical conditions. For instance, the Von Post scale, widely used to evaluate peat profiles, essentially measures the degree of humification. In our model, the first three soil organic matter (SOM) pools receive inputs exclusively from litter and represent a gradient of litter quality. The decomposition process, mediated by microbial activity and physical stress, results in the formation of an amorphous SOM compartment, which we term "peat." This approach aligns with other models, such as Yasso07 (Didion et al., 2014), where four SOM compartments associated with litter inputs decompose to form humus. In the YassoPeatland adaptation, this humus compartment is associated with the catotelm (Li et al., 2024). Similarly, the ESOM module in the SUSI peatland simulator follows this conceptual framework, where decomposition succession of litter-related compartments leads to peat formation (see reference: https://www.sciencedirect.com/science/article/pii/S0048969724053233).

Representing peat as a single conceptual compartment is a common practice in ecosystem models. For instance, the SUSI peatland simulator and models such as ORCHIDEE discretise peat into saturated and unsaturated layers. While we recognise the limitations of conceptual compartments, as noted in our discussion section, peat in real life is loosely defined. Chemically, it consists of varying proportions of microbial-derived compounds, cellulose-like compounds, lignin-like compounds, and others, with these proportions differing by peat type. Moving to a more analytically based compartmental structure might improve the representation of peat, but it would also reduce the model's practicality for many applications. Data on peat composition are far less common than data on peat thickness and carbon content, making a conceptual approach more viable for the moment.

**Changes in text due to R4C3:**

Line 679: Appendix A: Model sensitivity analysis…

Line 704: Appendix B: Soil physical changes…

**R4C4:**

*Line 80 in main text: ForSAFE has a mineral soil weathering component. This plays a key role in nutrient availability and forest productivity, but the dynamics are radically different on organic soils. How is this accounted for?*

**Answer R4C4:**

The weathering component in ForSAFE is based on the initial conditions of the soil's mineral fraction, further subdivided into specific mineral types, as described in the SAFE model. This component is

particularly relevant for nutrient availability as it provides a source of base cations. However, the differences in weathering dynamics between mineral and organic soils emerge naturally in the model due to the site-specific initial conditions.

In the peat-dominated site considered in this study, the mineral soil fraction is very small, comprising around 13% of the total soil composition, SOM is around 87% of the soil. While this fraction is incorporated into the model, its limited presence means that weathering has a negligible effect on nutrient availability due to a very small flux. Instead, nutrient dynamics in this site are predominantly governed by organic matter mineralisation processes, which are more representative of nutrient cycling in organic soils. This distinction ensures that the model accurately reflects the key drivers of nutrient availability and forest productivity under the specific conditions of the study site.

By explicitly representing both the mineral and organic soil components and their respective contributions to nutrient availability, the model captures the unique dynamics of organic soils while maintaining the flexibility to simulate sites with different soil compositions.

**R4C5:**

*Abstract: first line of abstract mentions greenhouse gases. For organic soils, also CH4 and N2O may play an important role: maybe emphasise in the abstract that you only focused on carbon dioxide?*

**Answer R4C5:**

We do include CH4, the site is just a small CH4 sink. To clarify this, we edited the abstract

**Changes in text due to R4C5:**

Line 16-18: Simulated carbon exchanges (a positive sign indicates gains and a negative sign indicates losses) were analysed considering different system boundaries (soil, ecosystem, and ecosystem plus the fate of harvested wood products, named ecosystem+HWP) using the net carbon balance (NCB) and the integrated carbon storage (ICS) metrics.

**R4C6:**

*Abstract: maybe give brief explanation on the metrics used, especially ICS? Also, clarify whether the negative or positive values indicate carbon loss or gain, conventions on this vary between metrics.*

**Answer R4C6:**

See "Changes in text due to R4C5"

**R4C7:**

*Line 37: C in gC often not in subscript*

**Answer R4C7:**

I agree that is not often in subscript but that has never sat well in my head. When information is added to a unit (in this paper and in many others) often subscripts are used. For example, when is cubic meter of water you often find $m^3_{water}$ or $m^3_w$ I have not found things as $m^3W$ so in order to be consistent I rather go with $g_C$.

**R4C8:**

*Lines 63 -- 64: add ref*

**Answer R4C8:**

We added Lehtonen et al. (2023) ([https://www.nature.com/articles/s41598-023-42315-7](https://www.nature.com/articles/s41598-023-42315-7))

**Changes in text due to R4C8:**

Line 68: … few decades (Lehtonen et al., 2023)

**R4C8:**

*Line 84: maybe intensively monitored instead of heavily?*

**Answer R4C8:**

Indeed, it is better.

**Changes in text due to R4C8:**

Line 101: For the first question, we compared model outputs to field measurements performed in an intensively monitored site using goodness of fit indicators …

**R4C9:**

*Line 103: is this a new development?*

**Answer R4C9:**

Not really, lateral flow was added by Zanchi et al. (2021). We have clarified this in a new paragraph, see answer to R4C1.

**R4C10:**

*Line 106: mineral weathering: how is this described if the peat layer is thick?*

**Answer R4C10:**

We have tried to answer to this concern in the answer R4C4.

**R4C11:**

*Lines 113 -- 114: add reference for decay constants*

**Answer R4C11:**

Here we just try to describe the model structure, we think that the supplementary material is detailed enough regarding the sources and values associated to the decay constant and other elements of the model

**R4C12:**

*Line 117: is there a better reference for the station?*

**Answer R4C12:**

We change it to Klemedtsson et al. (2015): (https://ui.adsabs.harvard.edu/abs/2015EGUGA..17.7461K/abstract)

**Changes in text due to R4C12:**

Line 141: ... at Skogaryd Research Station (Klemedtsson et al., 2015) in the southwest of Sweden

**R4C13:**

*Line 118: high nutrient content: nitrogen, or also others? Due to agricultural history, was it fertilised earlier?*

**Answer R4C13:**

We don't have information for every nutrient but CN ratios and NP ratios show high contents of nitrogen and phosphorus. Likely the agricultural history played an important role, but the deeper peat (1.5m to 2m) show high content of nitrogen and phosphorus suggesting nutrient content being associated with peat type.

**Changes in text due to R4C13:**

Line 143: This site experiences a hemiboreal climate, has nitrogen-rich peat soil ...

**R4C14:**

*Line 122: give also distances to the stations, coordinates hard to interpret*

**Answer R4C14:**

We have modified the text accordingly

**Changes in text due to R4C14:**

Line 159: ..., both located approximately 12km from the site

**R4C15:**

*Line 125: deposition of nitrogen?*

**Answer R4C15:**

Yes, among other elements for which deposition data is also available such as Cl and Na.

**R4C16:**

*Fig. 1 caption: is the difference between minima and maxima plotted, or minima and maxima explicitly? If not explicitly, why not?*

**Answer R4C16:**

We agree is unclear, we have a changed the caption

**Changes in text due to R4C16:**

Line 168: Figure 3. (a) Mean annual temperature (black line) and the range between the annual maximum and minimum temperatures (grey area).

**R4C17:**

*Line 140: is the assumption of only Norway spruce realistic?*

**Answer R4C17:**

We believe the assumption is reasonable, there was not any other species of tree in the site and the cycle of forest stand dominates carbon fluxes. However, we bring the limitations associated to not representing understory vegetation in the section (4.1) of discussion associated to representativeness.

**R4C18:**

*Line 140 -- 141: little snippets of information like this make it hard to follow what is old, what is new. Maybe restructure to have all model related stuff in one place*

**Answer R4C18:**

This section aims to provide information of the main characteristics of the scenario simulated in the model and relate those characteristics to information we have about the site and similar sites. For this case we reformulated to avoid discussing detailed parametrization, we reformulate the paragraph.

**Changes in text due to R4C18:**

Line 171-174: The simulated forest stand is assumed to consist entirely of Norway spruce. The modelled forest management replicated historical events at the site: spruce planting in 1951, a 72% tree biomass thinning in 1979, a 10% biomass loss in 2010 due to storm damage, and a 96% biomass removal in 2019 as part of a clear-cutting operation. Harvesting plays a crucial role in regulating carbon dynamics in such systems.

**R4C19:**

*Line 142: is three metres the depth of the peat?*

**Answer R4C19:**

Is an average because the site was a fen valley, therefore depth changes spatially being deeper at the center.

**R4C20:**

*Line 144: same properties for all layers: is this realistic? I would imagine different peat decomposition stages at different depths. Although I understand that this may be hard to verify*

**Answer R4C20:**

Indeed, giving the uncertainty about the conditions in 1950's and spatial (horizontal and vertical) variability exhibited by the site we consider that giving all layers the same bulk density, CN ratio and OM content is a reasonable simplification. We know we can't perfectly represent the site (digital twin) so the intention is to create a synthetic site that represents the main characteristics.

**R4C21:**

*Line 146: ref for the OM content*

**Answer R4C21:**

Modified in line 182

**Changes in text due to R4C21:**

Line 181-182: while soil organic matter content was set to 87% based on Meyer et al. (2013)

**R4C22:**

*Line 179: Is NCB same as net ecosystem exchange or productivity? What fluxes are included, also lateral?*

**Answer R4C22:**

NCB is calculated for every system boundary based on the definitions presented in the figure 3.

**R4C23:**

*Line 201, Fig. 2 caption: arrow a 1: this refers to leaching to runoff? Currently the arrow points upwards, and looks like offgassing from ditches*

**Answer R4C23:**

I agree but we try to clarify that in the text, we decided like this to simplify what is inflow (going down) and what is outflow (going up). In some cases, like leaching or harvested wood (8 in figure 3b), is not so logical, however we think it helps to quickly define inflows and outflows regardless of the true spatial characteristic of the flow (harvesting and leaching being lateral losses).

**R4C24:**

*Line 202, Fig. 2 caption: arrow a 3: does the soil itself take up atmospheric carbon dioxide? Litterfall is separately described. Also, SI says there are no processes that consume gaseous CO2 (line 243)*

**Answer R4C24:**

Carbon gas exchange is gradient-controlled, so in theory, it could flow both ways. In these particular conditions (drainage) the site emits CO2 and consumes CH4.

**Changes in text due to R4C24:**

Line 260-265: For all system boundaries, outputs are represented by negative fluxes and inputs by positive fluxes. The soil boundary includes inflows from litterfall and belowground autotrophic respiration, with outflows from leached carbon (e.g., dissolved organic carbon, $CO_2$, and $CH_4$). Soil-atmosphere carbon exchange is gradient-controlled and can act as either an input or output of gaseous carbon ($CO_2$ and $CH_4$). At the ecosystem boundary, photosynthesis is an inflow, while leaching, aboveground autotrophic respiration and harvested biomass are outflows. Soil-atmosphere exchange is also included. The ecosystem+HWP boundary accounts for the same fluxes as the ecosystem boundary, but harvested biomass is replaced by the decay of wood products.

**R4C25:**

*Line 202, arrow a4: would this not be outflow? Or does this mean release of CO2 into the soil headspace/matrix? I would guess that is rapidly lost through arrow 2*

**Answer R4C25:**

We are concerned about flows of carbon so $CO_2$ is not the only carbon flux. In this case this is carbon in litterfall being added to the soil

**R4C26:**

*Fig. 2 caption: some arrows stay the same between a, b and c, such as arrows 1 and 2. Some change (like arrows 3, 4 and 5). I would suggest keeping the meaning of each number the same, and having a few more numbers.*

**Answer R4C26:**

We think that is a good idea and the figure has been changed accordingly

**Changes in text due to R4C26:**

Line 258-261: System boundaries used in the study are (a) soil boundary, (b) ecosystem boundary, and (c) ecosystem + harvested wood products (HWP) boundary. Yellow arrows represent carbon outflows, and blue arrows represent carbon inflows: carbon leaching (arrow 1), soil-atmosphere carbon exchange (arrows 2 and 3), litterfall (arrow 4), belowground autotrophic respiration (arrow 5), aboveground autotrophic respiration (arrow 6), photosynthesis (arrow 7), harvested biomass (arrow 8), and outflows from the decay of HWPs (arrows 9, 10 and 11).

**R4C27:**

*Lines 587 -- 589, code availability. Would it be possible to make the model openly available? It is possible to get a persistent doi for a model release e.g. through Zenodo github integration. Do the contact details here refer to DE? Are there any guarantees that these work in ten years, and the model is still available? Even if the details would be permanent, I would prefer open availability of the model. I understand that this view might be not shared by everyone, but I would urge the authors to consider this possibility to promote open science.*

**Answer R4C27:**

We agree with the reviewer's concern, the reason of the current statement on code availability is associated to current work being performed in the code to include certain aspect we need to include for a future contribution of the effect of restoration. Therefore, we would like a final version of the code with the new improvements before releasing it.

**R4C28:**

*Fig. S1: The definition of the abbreviations is not clearly given in the text, it took me some looking to find them in Table S1 much lower down.*

**Answer R4C28:**

We have changed the figure caption to provide the missing information.

**Changes in text due to R4C28:**

Line 24-31 SPLMT: Figure S4. Scheme of carbon compartments within the model. The green boxes represent plant-related compartments, the brown boxes represent soil organic matter compartments, the grey boxes represent deadwood left after harvesting and harvested wood products (HWP), and the pink boxes represent $CO_2$ and $CH_4$ in the soil. The black arrows indicate carbon fluxes, with arrows not connected to any compartment representing fluxes leaving the

system. The subscripts denote specific carbon compartments: LC (labile central), LF (labile foliage), LR (labile root), LB (labile branch), LW (labile wood), TF (tissue foliage), TR (tissue root), TB (tissue branch), TW (tissue wood), HW (hardwood from harvest), HP (paper from harvest), HF (fuel from harvesting), TD (tissue deadwood), SE (soil easily decomposable compounds), SC (soil cellulose), SL (soil lignin), SP (soil peat) and SD (soil dissolved).

**R4C29:**

*Multiple points in manuscript and SI, for example line 157 in SI: linear decay of harvested wood. I assume first-order exponential decay is meant?*

**Answer R4C29:**

We correct accordingly throughout the manuscript. Some example below.

**Changes in text due to R4C29:**

Line 563:... that decomposes following first-order exponential decay

Line 572: A description of decomposition based on first-order exponential decay is not adequate to capture non-linear

**R4C30:**

*SI, Line 295: modified how exactly? Nitrogen dynamics on organic soil are very different from those on mineral soil*

**Answer R4C30:**

We tried to clarify this in the answer R4C1 by improving the first paragraphs of section 1 of the supplementary material to provide more clarity to the changes performed to the model for this contribution.

**R4C31:**

*SI, section 2.2: I initially was confused by the Harvest removal and Harvest intensity parameters, maybe these could be clarified in the text?*

**Answer R4C31:**

We have added a sentence to clarify.

**Changes in text due to R4C31:**

Line 427-429: The harvest intensity parameter ($HI$) specifies the proportion of the forest stand that is cut during harvesting, whereas the harvest removal parameter ($HR$) defines the fraction of the harvested material that is removed from the site